# Products, Abstractions and Inclusions of Causal Spaces

**Simon Buchholz**[*1]          **Junhyung Park**[*1]          **Bernhard Schölkopf**[1]

[1]Empirical Inference Department, Max Planck Institute for Intelligent Systems, Tübingen, Germany

## Abstract

Causal spaces have recently been introduced as a measure-theoretic framework to encode the notion of causality. While it has some advantages over established frameworks, such as structural causal models, the theory is so far only developed for single causal spaces. In many mathematical theories, not least the theory of probability spaces of which causal spaces are a direct extension, combinations of objects and maps between objects form a central part. In this paper, taking inspiration from such objects in probability theory, we propose the definitions of products of causal spaces, as well as (stochastic) transformations between causal spaces. In the context of causality, these quantities can be given direct semantic interpretations as causally independent components, abstractions and extensions.

## 1  INTRODUCTION

Mathematical modelling of the world allows us to represent and analyse real-life situations in a quantitative manner. Depending on the aspects that one is interested in, different mathematical tools are chosen for the modelling; for example, to model how a system evolves over time, a system of differential equations can be used, and to model the randomness of events, one can use probability theory. Another aspect of the world which researchers are increasingly more interested in modelling is causality [Woodward, 2005, Russo, 2010, Illari et al., 2011, Pearl and Mackenzie, 2018], and this is the focus of our work.

For the mathematical frameworks used in modelling, there are always ways to analyse multiple structures in a coherent manner. For example, in vector spaces, we have the notions of subspaces, product spaces and maps between vector spaces. In probability spaces, we have the notions of subspaces and restrictions, product spaces and measurable maps and transition probability kernels between spaces.

In this work, we consider a modelling of the world through a recently proposed framework called *causal spaces* [Park et al., 2023]. Causal spaces are a direct extension of probability spaces to encode causal information, and as such, are rigorously grounded in measure-theory. While they have some advantages over existing frameworks (e.g. structural causal models, or SCMs), such as the fact that they can easily encode cycles and continuous-time stochastic processes that are notoriously problematic in SCMs [Halpern, 2000, Bongers et al., 2021], the theory of causal spaces is still in its infancy. In particular, Park et al. [2023] only consider the development of *single* causal spaces, and omit the discussion of construction of new causal spaces from existing ones or maps between causal spaces. The latter is of particular interest to researchers in causality for the purpose of *abstraction* (see related works in Section 1.1). When systems, humans or animals perceive the world, they consider different levels of detail depending on their ability to perceive and retain information and their level of interest. It is therefore crucial to connect the mathematical representations at varying levels of granularity in a coherent way.

In probability spaces, such notions are well-established. Product measures give rise to independent random variables, and measurable maps and probability kernels between probability spaces give rise to pushforward measures, which can be interpreted as abstractions or inclusions. Based on these concepts, and using the fact that causal spaces are a direct extension of probability spaces, we develop the notions of *product causal spaces* and *causal transformations*.

---
*Equal Contribution.

## 1.1 RELATED WORKS

The theory of causality has two dominant strands [Imbens, 2019], one based on SCMs [Pearl, 2009, Peters et al., 2017] and another based on potential outcomes [Hernàn and Robins, 2020, Imbens and Rubin, 2015]. Since concepts such as abstractions and connected components attract much more attention in the SCM community than in the potential outcomes community, we focus on comparisons with the SCM framework.

Seminal works on causal abstraction with SCMs are Rubenstein et al. [2017] and Beckers and Halpern [2019], where the notions of exact transformations (to be further discussed in Section 5), uniform transformations, abstractions, strong abstractions and constructive abstractions are proposed. Beckers et al. [2020] then relax these to an approximate notion. Massidda et al. [2023] extended the notions to soft interventions, and Zečević et al. [2023] to continually updated abstractions. Causal feature learning is a closely related approach, that also aims to learn higher level features [Chalupka et al., 2015, 2016, 2017] There are also approaches based on category theory [Rischel and Weichwald, 2021, Otsuka and Saigo, 2022, 2023] and probabilistic logic [Ibeling and Icard, 2023], all grounded in SCMs; see [Zennaro, 2022] for a review.

The notion of causal abstraction in the SCM framework has found applications in interpretations of neural networks [Geiger et al., 2021, 2023] as well as solving causal inference tasks (identification, estimation and sampling) at different levels of granularity with neural networks [Xia and Bareinboim, 2024]. Moreover, Zennaro et al. [2023] proposed a way of *learning* an abstraction from partial information about the abstraction, and demonstrates an application of causal abstraction in the SCM framework in the context of electric vehicle battery manufacturing and Kekić et al. [2023] learn an abstraction that explains a specific target.

## 1.2 PAPER ORGANISATION

The rest of this paper is structured as follows. We first discuss the key notions from the theory of causal spaces in Section 2. Then we introduce the extension of product causal spaces in Section 3 followed by our definitions of transformations of causal spaces in Section 4. We put our definitions into context by comparing carefully to related works in Section 5. In Section 6, we then show various properties of our transformations, in particular for the subclass of abstractions.

## 2 PRELIMINARIES & NOTATIONS

We take a probability space as a starting point, namely, a triple $(\Omega, \mathcal{H}, \mathbb{P})$; for a comprehensive introduction, see, for example, [Cinlar, 2011, Durrett, 2019]. Following [Park et al., 2023], we additionally insist that $\mathbb{P}$ is defined over the product measurable space $(\Omega, \mathcal{H}) = \otimes_{t \in T}(E_t, \mathcal{E}_t)$ with $(E_t, \mathcal{E}_t)$ being the same standard measurable space if $T$ is uncountable. Denote by $\mathcal{P}(T)$ the power set of $T$, and for $S \in \mathcal{P}(T)$, we denote by $\mathcal{H}_S$ the sub-$\sigma$-algebra of $\mathcal{H} = \otimes_{t \in T} \mathcal{E}_t$ generated by measurable rectangles $\times_{t \in T} A_t$, where $A_t \in \mathcal{E}_t$ differs from $E_t$ only for $t \in S$ and finitely many $t$. In particular, $\mathcal{H}_\emptyset = \{\emptyset, \Omega\}$ is the trivial sub-$\sigma$-algebra of $\Omega = \times_{t \in T} E_t$. Also, we denote by $\Omega_S$ the subspace $\times_{s \in S} E_s$ of $\Omega = \times_{t \in T} E_t$, and for $T \supseteq S \supseteq U$, we let $\pi_{SU}$ denote the natural projection from $\Omega_S$ onto $\Omega_U$ and we use the shorthand $\pi_S = \pi_{TS}$. We also write $\omega_S = \pi_S \omega = (\omega_s)_{s \in S}$. We write $[d] = \{1, \ldots, d\}$ and $f_* \mathbb{P}$ denotes the pushforward measure along the map $f$, namely, $f^* \mathbb{P}(A) = \mathbb{P}(f^{-1}(A))$.

We first recall the definition of causal spaces, as given in Park et al. [2023].

**Definition 2.1** (Causal Spaces, [Park et al., 2023, Definition 2.2]). A *causal space* is defined as the quadruple $(\Omega, \mathcal{H}, \mathbb{P}, \mathbb{K})$, where $(\Omega, \mathcal{H}, \mathbb{P}) = (\times_{t \in T} E_t, \otimes_{t \in T} \mathcal{E}_t, \mathbb{P})$ is a probability space and $\mathbb{K} = \{K_S : S \in \mathcal{P}(T)\}$, called the *causal mechanism*, is a collection of transition probability kernels $K_S$ from $(\Omega, \mathcal{H}_S)$ into $(\Omega, \mathcal{H})$, called the *causal kernel on $\mathcal{H}_S$*, that satisfy the following axioms:

(i) for all $A \in \mathcal{H}$ and $\omega \in \Omega$, we have $K_\emptyset(\omega, A) = \mathbb{P}(A)$;

(ii) for all $\omega \in \Omega$, and events $A \in \mathcal{H}_S$ and $B \in \mathcal{H}$, we have $K_S(\omega, A \cap B) = \mathbf{1}_A(\omega) K_S(\omega, B) = \delta_\omega(A) K_S(\omega, B)$.

The causal kernels $K_S$ can be defined equivalently as maps from $(\Omega_S, \mathcal{H}_S)$ to $(\Omega, \mathcal{H})$ and it will be convenient to use this viewpoint occasionally in the following (this was also used implicitly in Park et al. [2023]). This observation is explained in Appendix C.

Next, we recall the definition of *interventions*, which is *the* central concept in any theory of causality.

**Definition 2.2** (Interventions, [Park et al., 2023, Definition 2.3]). Let $(\Omega, \mathcal{H}, \mathbb{P}, \mathbb{K}) = (\times_{t \in T} E_t, \otimes_{t \in T} \mathcal{E}_t, \mathbb{P}, \mathbb{K})$ be a causal space, $U \subseteq T$ a subset, $\mathbb{Q}$ a probability measure on $(\Omega, \mathcal{H}_U)$ and $\mathbb{L} = \{L_V : V \in \mathcal{P}(U)\}$ a causal mechanism on $(\Omega, \mathcal{H}_U, \mathbb{Q})$. An *intervention on $\mathcal{H}_U$ via $(\mathbb{Q}, \mathbb{L})$* is a new causal space $(\Omega, \mathcal{H}, \mathbb{P}^{\mathrm{do}(U,\mathbb{Q})}, \mathbb{K}^{\mathrm{do}(U,\mathbb{Q},\mathbb{L})})$, where the *intervention measure* $\mathbb{P}^{\mathrm{do}(U,\mathbb{Q})}$ is a probability measure on $(\Omega, \mathcal{H})$ defined, for $A \in \mathcal{H}$, by

$$\mathbb{P}^{\mathrm{do}(U,\mathbb{Q})}(A) = \int \mathbb{Q}(d\omega_U) K_U(\omega_U, A)$$

and $\mathbb{K}^{\mathrm{do}(U,\mathbb{Q},\mathbb{L})} = \{K_S^{\mathrm{do}(U,\mathbb{Q},\mathbb{L})} : S \subseteq T\}$ is the *intervention causal mechanism* whose *intervention causal kernels* are

$$K_S^{\mathrm{do}(U,\mathbb{Q},\mathbb{L})}(\omega_S, A)$$

$$= \int L_{S \cap U}(\omega_{S \cap U}, d\omega'_U) K_{S \cup U}((\omega_{S \setminus U}, \omega'_U), A).$$

We also recall the definition of *causal effect*.

**Definition 2.3** (Causal Effects, [Park et al., 2023, Definition B.1]). Let $(\Omega, \mathcal{H}, \mathbb{P}, \mathbb{K}) = (\times_{t \in T} E_t, \otimes_{t \in T} \mathcal{E}_t, \mathbb{P}, \mathbb{K})$ be a causal space, $U \subseteq T$ a subset, $A \in \mathcal{H}$ an event and $\mathcal{F}$ a sub-$\sigma$-algebra of $\mathcal{H}$ (not necessarily of the form $\mathcal{H}_S$ for some $S \in \mathcal{P}(T)$).

(i) If $K_S(\omega, A) = K_{S \setminus U}(\omega, A)$ for all $S \in \mathcal{P}(T)$ and all $\omega \in \Omega$, then we say that $\mathcal{H}_U$ has *no causal effect* on $A$, or that $\mathcal{H}_U$ is *non-causal* to $A$.

We say that $\mathcal{H}_U$ has *no causal effect* on $\mathcal{F}$, or that $\mathcal{H}_U$ is *non-causal* to $\mathcal{F}$, if, for all $A \in \mathcal{F}$, the $\sigma$-algebra $\mathcal{H}_U$ has no causal effect on $A$.

(ii) If there exists $\omega \in \Omega$ such that $K_U(\omega, A) \neq \mathbb{P}(A)$, then we say that $\mathcal{H}_U$ has an *active causal effect* on $A$, or that $\mathcal{H}_U$ is *actively causal* to $A$.

We say that $\mathcal{H}_U$ has an *active causal effect* on $\mathcal{F}$, or that $\mathcal{H}_U$ is *actively causal* to $\mathcal{F}$, if $\mathcal{H}_U$ has an active causal effect on some $A \in \mathcal{F}$.

(iii) Otherwise, we say that $\mathcal{H}_U$ has a *dormant causal effect* on $A$, or that $\mathcal{H}_U$ is *dormantly causal* to $A$.

We say that $\mathcal{H}_U$ has a *dormant causal effect* on $\mathcal{F}$, or that $\mathcal{H}_U$ is *dormantly causal* to $\mathcal{F}$, if $\mathcal{H}_U$ does not have an active causal effect on any event in $\mathcal{F}$ and there exists $A \in \mathcal{F}$ on which $\mathcal{H}_U$ has a dormant causal effect.

Finally, we recall the definition of *sources*, which allows us to connect the causal kernels to the probability measure $\mathbb{P}$. For a sub-$\sigma$-algebra $\mathcal{F}$ of $\mathcal{H}$, we denote the *conditional probability* of an event $A \in \mathcal{H}$ given $\mathcal{F}$ by $\mathbb{P}_{\mathcal{F}}$.

**Definition 2.4** (Sources, [Park et al., 2023, Definition D.1]). Let $(\Omega, \mathcal{H}, \mathbb{P}, \mathbb{K}) = (\times_{t \in T} E_t, \otimes_{t \in T} \mathcal{E}_t, \mathbb{P}, \mathbb{K})$ be a causal space, $U \subseteq T$ a subset, $A \in \mathcal{H}$ an event and $\mathcal{F}$ a sub-$\sigma$-algebra of $\mathcal{H}$. We say that $\mathcal{H}_U$ is a *(local) source* of $A$ if $K_U(\cdot, A)$ is a version of the conditional probability $\mathbb{P}_{\mathcal{H}_U}(A)$. We say that $\mathcal{H}_U$ is a *(local) source* of $\mathcal{F}$ if $\mathcal{H}_U$ is a source of all $A \in \mathcal{F}$. We say that $\mathcal{H}_U$ is a *global source* of the causal space if $\mathcal{H}_U$ is a source of all $A \in \mathcal{H}$.

# 3 PRODUCT CAUSAL SPACES AND CAUSAL INDEPENDENCE

We first give the definition of the product of causal kernels, and the product of causal spaces. This constitutes the simplest way of constructing new causal spaces from existing ones.

**Definition 3.1** (Product Causal Spaces). Suppose $\mathcal{C}^1 = (\Omega^1, \mathcal{H}^1, \mathbb{P}^1, \mathbb{K}^1)$ and $\mathcal{C}^2 = (\Omega^2, \mathcal{H}^2, \mathbb{P}^2, \mathbb{K}^2)$ with $\Omega^1 = \times_{t \in T^1} E_t$ and $\Omega^2 = \times_{t \in T^2} E_t$ are two causal spaces. For

all $S^1 \subseteq T^1$ and $S^2 \subseteq T^2$, and for a pair of causal kernels $K_{S^1}^1 \in \mathbb{K}^1$ and $K_{S^2}^2 \in \mathbb{K}^2$, we define the *product causal kernel* $K_{S^1}^1 \otimes K_{S^2}^2$, for $\omega = (\omega_1, \omega_2) \in \Omega_{S^1}^1 \times \Omega_{S^2}^2$ and events $A_1 \in \mathcal{H}^1$ and $A_2 \in \mathcal{H}^2$, by

$$K_{S^1}^1 \otimes K_{S^2}^2(\omega, A_1 \times A_2) = K_{S^1}^1(\omega_1, A_1) K_{S^2}^2(\omega_2, A_2).$$

This can then be extended to all of $\mathcal{H}^1 \otimes \mathcal{H}^2$ since the rectangles $A_1 \times A_2$ with $A_1 \in \mathcal{H}^1$ and $A_2 \in \mathcal{H}^2$ generate $\mathcal{H}^1 \otimes \mathcal{H}^2$. Then we define the *product causal space*

$$\mathcal{C}^1 \otimes \mathcal{C}^2 = (\Omega^1 \times \Omega^2, \mathcal{H}^1 \otimes \mathcal{H}^2, \mathbb{P}^1 \otimes \mathbb{P}^2, \mathbb{K}^1 \otimes \mathbb{K}^2)$$

where the product causal mechanism $\mathbb{K}^1 \otimes \mathbb{K}^2$ is the unique family of kernels of the form $(K^1 \otimes K^2)_{S^1 \cup S^2} = K_{S^1}^1 \otimes K_{S^2}^2$ for $S^1 \subseteq T^1$ and $S^2 \subseteq T^2$.

We first check that this procedure indeed produces a valid causal space.

**Lemma 3.2** (Products of Causal Spaces are Causal Spaces). *The product causal space $\mathcal{C}^1 \otimes \mathcal{C}^2$ as defined in Definition 3.1 is a causal space.*

The proof of this lemma can be found in Appendix B.1.

Note that it is only for the sake of simplicity of presentation that we presented the notion of products only for two probability spaces. Indeed, we can easily extend the definition to arbitrary products of causal kernels and causal spaces, just like it is possible for products of probability spaces.

When we take a product of causal spaces, the corresponding components in the resulting causal space do not have a causal effect on each other, as the following result shows.

**Lemma 3.3** (Causal Effects in Product Spaces). *Suppose $\mathcal{C}^1 = (\Omega^1, \mathcal{H}^1, \mathbb{P}^1, \mathbb{K}^1)$ and $\mathcal{C}^2 = (\Omega^2, \mathcal{H}^2, \mathbb{P}^2, \mathbb{K}^2)$ with $\Omega^1 = \times_{t \in T^1} E_t$ and $\Omega^2 = \times_{t \in T^2} E_t$ are two causal spaces. Then in $\mathcal{C}^1 \otimes \mathcal{C}^2$,*

(i) *$\mathcal{H}_{T^1}$ has no causal effect on $\mathcal{H}_{T^2}$, and $\mathcal{H}_{T^2}$ has no causal effect on $\mathcal{H}_{T^1}$;*

(ii) *$\mathcal{H}_{T^1}$ and $\mathcal{H}_{T^2}$ are (local) sources of each other.*

The proof of this Lemma is in Appendix B.1.

Product causal spaces are analogous to *connected components* in graphical models – see, for example, [Sadeghi and Soo, 2023].

## 3.1 CAUSAL INDEPENDENCE

Recall that, in probability spaces, two events $A$ and $B$ are *independent* with respect to the measure $\mathbb{P}$ if $\mathbb{P}(A \cap B) = \mathbb{P}(A)\mathbb{P}(B)$, i.e. the probability measure is the product measure. Moreover, two $\sigma$-algebras are independent if each pair

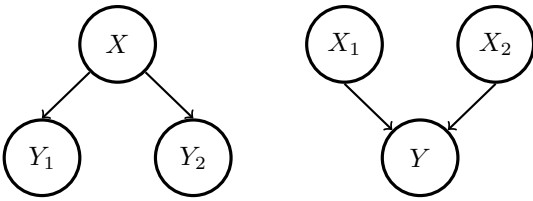

Figure 1: Graphs of SCMs in Example 3.5.

of events from the two $\sigma$-algebras are independent[1]. Similarly, for a sub-$\sigma$-algebra $\mathcal{F}$ of $\mathcal{H}$, two events $A$ and $B$ are *conditionally independent given* $\mathcal{F}$ if $\mathbb{P}_{\mathcal{F}}(A \cap B) = \mathbb{P}_{\mathcal{F}}(A)\mathbb{P}_{\mathcal{F}}(B)$ almost surely, and two $\sigma$-algebras are conditionally independent given $\mathcal{F}$ if each pair of events from the two $\sigma$-algebras are conditionally independent given $\mathcal{F}$.

Now we give a definition of causal independence that is analogous to conditional independence.

**Definition 3.4** (Causal Independence). Consider a causal space $\mathcal{C} = (\Omega = \times_{t \in T} E_t, \mathcal{H} = \otimes_{t \in T} \mathcal{E}_t, \mathbb{P}, \mathbb{K})$. Then for $U \subseteq T$, two events $A, B \in \mathcal{H}$ are *causally independent on* $\mathcal{H}_U$ if, for all $\omega \in \Omega$,

$$K_U(\omega, A \cap B) = K_U(\omega, A)K_U(\omega, B).$$

We say that two sub-$\sigma$-algebras $\mathcal{F}_1$ and $\mathcal{F}_2$ are *causally independent on* $\mathcal{H}_U$ if each pair of events from $\mathcal{F}_1$ and $\mathcal{F}_2$ are causally independent on $\mathcal{H}_U$.

Semantically, causal independence should be interpreted as follows: if $A$ and $B$ are causally independent on $\mathcal{H}_U$, then they are independent once an intervention has been carried out on $\mathcal{H}_U$. Note also that causal independence is really about the causal kernels, and has nothing to do with the probability measure $\mathbb{P}$ of the causal space. Indeed, it is possible for $A$ and $B$ to be causally independent but not probabilistically independent, or causally independent but not conditionally independent, or vice versa. Let us illustrate with the following simple examples.

**Example 3.5** (Causal Independence). *(i) Consider three variables $X$, $Y_1$ and $Y_2$ related through the equations*

$$X = N, \quad Y_1 = X + U_1, \quad Y_2 = X + U_2,$$

*where $N$, $U_1$ and $U_2$ are standard normal variables (see Figure 1 left). We denote by $\mathbb{P}$ their joint distribution on $\mathbb{R}^3$, and we identify this SCM with the causal space $(\mathbb{R}^3, \mathcal{B}(\mathbb{R}^3), \mathbb{P}, \mathbb{K})^2$, where $\mathbb{K}$ is obtained via the above structural equations. Then it is clear to see that $Y_1$ and $Y_2$ are causally independent on $\mathcal{H}_X$, since, for*

---

[1]Many authors take the view that the notion of independence is truly where probability theory starts, as a distinct theory from measure theory [Cinlar, 2011, p.82, Section II.5].

[2]Here, $\mathcal{B}$ represents the Borel $\sigma$-algebra.

*every $x$, and $A, B \in \mathcal{B}(\mathbb{R})$, $K_X(x, \{Y_1 \in A, Y_2 \in B\})$ is bivariate-normally distributed with mean $(x, x)$ and identity covariance matrix, and so*

$$K_X(x, \{Y_1 \in A, Y_2 \in B\})$$
$$= K_X(x, \{Y_1 \in A\})K_X(x, \{Y_2 \in B\}).$$

*By the same reasoning, $Y_1$ and $Y_2$ are conditionally independent given $\mathcal{H}_X$. However, it is clear that they are unconditionally dependent, because they both depend on the value of $X$.*

*(ii) Now consider three variables $X_1$, $X_2$ and $Y$ related through the equations*

$$X_1 = N_1, \quad X_2 = N_2, \quad Y = X_1 + X_2 + U$$

*where $N_1$, $N_2$ and $U$ are standard normal variables (see Figure 1 right). We denote by $\mathbb{P}$ their joint distribution on $\mathbb{R}^3$, and we identify this SCM with the causal space $(\mathbb{R}^3, \mathcal{B}(\mathbb{R}^3), \mathbb{P}, \mathbb{K})$, where $\mathbb{K}$ is obtained via the above structural equations. Then it is clear that $X_1$ and $X_2$ are probabilistically independent. They are also causally independent on $\mathcal{H}_Y$, since, for any $A, B \in \mathcal{B}(\mathbb{R})$,*

$$K_Y(y, \{X_1 \in A, X_2 \in B\}) = \mathbb{P}(X_1 \in A, X_2 \in B)$$
$$= \mathbb{P}(X_1 \in A)\mathbb{P}(X_2 \in B).$$

*However, it is clear that they are conditionally dependent given $\mathcal{H}_Y$.*

Again, causal independence is only defined for two $\sigma$-algebras for the sake of notational convenience; it can easily be extended to arbitrary collections of $\sigma$-algebra.

# 4  TRANSFORMATIONS OF CAUSAL SPACES

Consider causal spaces $\mathcal{C}^1 = (\Omega^1, \mathcal{H}^1, \mathbb{P}^1, \mathbb{K}^1)$ and $\mathcal{C}^2 = (\Omega^2, \mathcal{H}^2, \mathbb{P}^2, \mathbb{K}^2)$ with $\Omega^1 = \times_{t \in T^1} E_t$ and $\Omega^2 = \times_{t \in T^2} E_t$. We want to define transformations between causal spaces $\mathcal{C}^1$ and $\mathcal{C}^2$. These transformations shall, on the one hand, preserve aspects of the causal structure, i.e., the spaces $\mathcal{C}^1$ and $\mathcal{C}^2$ shall still describe essentially the same system. On the other hand, they shall be flexible so that different types of mappings between causal spaces can be captured.

We focus on transformations that preserve individual variables or combine them in a meaningful way. This relation will be encoded by a map $\rho : T^1 \to T^2$, which can be interpreted as encoding the fact that $S \subseteq T^2$ depends only on the variables indexed by $\rho^{-1}(S)$. Deterministic maps are not sufficiently expressive for our purposes and we therefore focus on stochastic maps, i.e., on probability kernels from measurable spaces $(\Omega^1, \mathcal{H}^1)$ to $(\Omega^2, \mathcal{H}^2)$.

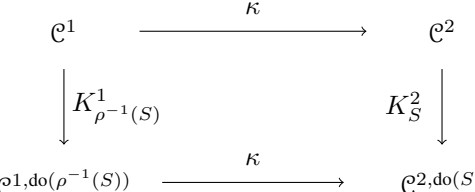

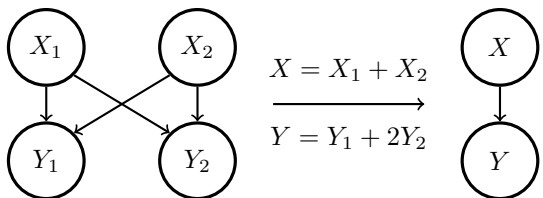

Figure 2: Interventional Consistency Definition 4.2 Equation (2) – intervention and transformation commute.

Figure 3: Abstraction of SCMs in Example 4.3.

**Definition 4.1** (Admissible Maps). Suppose that $\kappa : \Omega^1 \times \mathcal{H}^2 \to [0,1]$ is a probability kernel and $\rho : T^1 \to T^2$ is a map. Then we call the pair $(\kappa, \rho)$ *admissible* if $\kappa(\cdot, A)$ is $\mathcal{H}^1_{\rho^{-1}(S)}$ measurable for all $S \subset \rho(T^1)$ and $A \in \mathcal{H}^2_S$.

One difference between probability theory and causality seems to be that the latter requires the notion of variables (equivalently a product structure of the underlying space) that define entities that can be intervened upon. For a meaningful relation between two causal spaces, their interventions should be related, which requires some preservation of variables. The definition of admissible maps captures the fact that variables from $\rho^{-1}(S)$ are combined to form a new summary collection of variables indexed by $S$.

We now require maps between causal spaces to respect the distributional and interventional structure in the following sense.

**Definition 4.2** (Causal Transformations). A *transformation of causal spaces*, or a *causal transformation*, $\varphi : \mathcal{C}^1 \to \mathcal{C}^2$ is an admissible pair $\varphi = (\kappa, \rho)$ satisfying the following two properties.

(i) The map satisfies *distributional consistency*, i.e., for $A \in \mathcal{H}^2$

$$\int \mathbb{P}^1(\mathrm{d}\omega)\,\kappa(\omega, A) = \mathbb{P}^2(A). \tag{1}$$

(ii) The map satisfies *interventional consistency*, i.e., for all $A \in \mathcal{H}^2_{\rho(T^1)}$, $S \subset \rho(T^1)$, and $\omega \in \Omega^1$ the following holds

$$\int K^1_{\rho^{-1}(S)}(\omega, \mathrm{d}\omega')\kappa(\omega', A)$$
$$= \int \kappa(\omega, \mathrm{d}\omega')K^2_S(\omega', A). \tag{2}$$

Interventional consistency requires that interventions and causal transformations commute, i.e., the result of first intervening and then applying the transformation is the same as intervening on the target after the transformation – see Figure 2. We emphasise that in Definition 4.1 and 4.2 we do not prescribe conditions for added components indexed by $T^2 \backslash \rho(T^1)$. Further, we remark that as a special case, we can

accommodate deterministic maps $f : \Omega_1 \to \Omega_2$ by considering the associated probability kernel $\kappa_f(\omega, A) = \mathbf{1}_A(f(\omega))$. In this case, the admissibility condition reduces to the statement that $\pi_S \circ f$ is measurable with respect to $\mathcal{H}^1_{\rho^{-1}(S)}$ for all $S \subset \rho(T^1)$ and distributional consistency becomes, for $A \in \mathcal{H}^2$,

$$\mathbb{P}^2(A) = \int \mathbb{P}^1(\mathrm{d}\omega)\kappa(\omega, A)$$
$$= \int \mathbb{P}^1(\mathrm{d}\omega)\mathbf{1}_A(f(\omega))$$
$$= \mathbb{P}^1(f^{-1}(A))$$

so $f_*\mathbb{P}^1 = \mathbb{P}^2$ is the pushforward measure of $\mathbb{P}^1$ along $f$. Interventional consistency then reads

$$K^1_{\rho^{-1}(S)}(\omega, f^{-1}(A)) = K^2_S(f(\omega), A) \tag{3}$$

for all $A \in \mathcal{H}^2_{\rho(T^1)}$, $S \subset \rho(T^1)$, and $\omega \in \Omega^1$. Alternatively this can be expressed as

$$f_*K^1_{\rho^{-1}(S)}(\omega, A) = K^2_S(f(\omega), A).$$

where the push-forward acts on the measure defined by the probability kernel for some fixed $\omega$. Henceforth, with a slight abuse of notation, we denote deterministic maps by $(f, \rho)$ without resorting to the associated probability kernel.

## 4.1 EXAMPLES

Let us provide four prototypical examples of maps between causal spaces that are covered by this definition. Here we resort to the language of SCMs because they are a convenient framework that fits into causal spaces.

**Example 4.3** (Abstraction). *We consider four variables $X_1$, $X_2$, $Y_1$, and $Y_2$ which are related through the equations*

$$X_1 = N_1, \qquad\qquad X_2 = N_2,$$
$$Y_1 = 3X_1 + X_2 + U_1, \qquad Y_2 = X_2 + U_2$$

*where $U_1$, $U_2$, $N_1$, $N_2$ are independent standard normal variables. We denote by $\mathbb{P}$ their joint distribution on $\mathbb{R}^4$. Consider*

$$X = N, \qquad Y = 3X + U$$

*where $N \sim N(0,2)$ and $U \sim N(0,5)$. Denote their joint distribution on $\mathbb{R}^2$ by $\mathbb{Q}$. We identify the two SCMs with causal spaces $(\mathbb{R}^4, \mathcal{B}(\mathbb{R}^4), \mathbb{P}, \mathbb{K})$ and $(\mathbb{R}^2, \mathcal{B}(\mathbb{R}^2), \mathbb{Q}, \mathbb{L})$ as explained in [Park et al., 2023, Section 3.1].*

*Consider the deterministic map $f : \mathbb{R}^4 \to \mathbb{R}^2$ given by $f(x_1, x_2, y_1, y_2) = (x_1 + x_2, y_1 + 2y_2)$ and the map $\rho : [4] \to [2]$ given by $\rho(1) = \rho(2) = 1$, $\rho(3) = \rho(4) = 2$. Clearly, the pair $(f, \rho)$ is admissible as defined in Definition 4.1. It can be checked that*

$$\mathbb{E}[(X_1 + X_2)^2] = 2 = \mathbb{E}[X^2]$$
$$\mathbb{E}[(Y_1 + 2Y_2)^2] = 23 = \mathbb{E}[Y^2]$$
$$\mathbb{E}[(X_1 + X_2)(Y_1 + 2Y_2)] = 6 = \mathbb{E}[XY]$$

*which implies that $f_*\mathbb{P} = \mathbb{Q}$ because both distributions are centered Gaussian and their covariance matrices agree.*

*The non-trivial causal consistency relation (2) concerns interventions on $\{X_1, X_2\}$ and $X$ and on $\{Y_1, Y_2\}$ and $Y$. Note that*

$$K_{\{X_1, X_2\}}((x_1, x_2, y_1, y_2), \cdot)$$
$$= \delta_{(x_1, x_2)} \otimes N\left(\begin{pmatrix} 3x_1 + x_2 \\ x_2 \end{pmatrix}, \mathrm{Id}_2\right).$$

*Then we obtain*

$$f_*K_{\{X_1, X_2\}}((x_1, x_2, y_1, y_2), \cdot)$$
$$= \delta_{x_1 + x_2} \otimes N(3x_1 + 3x_2, 5).$$

*On the other hand, we find*

$$L_X((x, y), \cdot) = \delta_x \otimes N(3x, 5)$$
$$\Rightarrow L_X(f(x_1, x_2, y_1, y_2), \cdot) = \delta_{x_1 + x_2} \otimes N(3x_1 + 3x_2, 5)$$

*so that we see that (3) holds in this case. Similarly, we obtain*

$$K_{\{Y_1, Y_2\}}((x_1, x_2, y_1, y_2), \cdot) = N(0, \mathrm{Id}_1) \otimes \delta_{(y_1, y_2)},$$
$$L_Y((x, y), \cdot) = N(0, 2) \otimes \delta_y.$$

*We again find*

$$f_*K_{\{Y_1, Y_2\}}((x_1, x_2, y_1, y_2), \cdot)$$
$$= L_Y((x_1 + x_2, y_1 + 2y_2), \cdot).$$

This example shows abstraction, i.e., we obtain a transformation to a more coarse-grained view of the system. Note that interventional consistency is quite restrictive to satisfy, e.g., here it is crucial that all distributions are Gaussian so that all conditional distributions are also Gaussian.

Next, we consider an example that allows us to embed a causal space in a larger space that adds an independent disjoint system. For this, we make use of the definition of product causal spaces (Definition 3.1). In this case, the transformation is stochastic.

$$\mathcal{C}^1 \xrightarrow[\kappa(\omega, \cdot) = \delta_\omega \otimes \mathbb{P}^2]{\rho(t) = t} \mathcal{C}^1 \otimes \mathcal{C}^2$$

Figure 4: Inclusions of component causal spaces into the product (Example 4.4).

**Example 4.4** (Inclusion). *Let $\mathcal{C}^1 = (\Omega^1, \mathcal{H}^1, \mathbb{P}^1, \mathbb{K}^1)$ and $\mathcal{C}^2 = (\Omega^2, \mathcal{H}^2, \mathbb{P}^2, \mathbb{K}^2)$ be two causal spaces, with $\Omega^1 = \times_{t \in T^1} E_t$ and $\Omega^2 = \times_{t \in T^2} E_t$. We define an inclusion map $(\kappa, \rho) : \mathcal{C}^1 \to \mathcal{C}^1 \otimes \mathcal{C}^2$ by considering $\rho(t) = t$ for $t \in T^1$ and $\kappa(\omega, \cdot) = \delta_\omega \otimes \mathbb{P}^2$ (see Figure 4). This pair is clearly admissible and satisfies distributional consistency:*

$$\int \mathbb{P}^1(d\omega)\kappa(\omega, A_1 \times A_2) = \int \mathbb{P}^1(d\omega)\mathbf{1}_{A_1}(\omega)\mathbb{P}^2(A_2)$$
$$= \mathbb{P}^1(A_1)\mathbb{P}^2(A_2).$$

*Moreover, for any $S \subset T^1$, $\omega \in \Omega^1$, $A_1 \in \mathcal{H}^1$ and $A_2 \in \mathcal{H}^2$, we have*

$$\int K_S^1(\omega, d\omega')\kappa(\omega', A_1 \times A_2) = \mathbb{P}^2(A_2)K_S^1(\omega, A_1)$$

*and also,*

$$\int \kappa(\omega, d\omega_1' d\omega_2')K_S^1 \otimes K_\emptyset^2((\omega_1', \omega_2'), A_1 \times A_2)$$
$$= \int \kappa(\omega, d\omega_1' d\omega_2')K_S^1(\omega_1', A_1)K_\emptyset^2(\omega_2', A_2)$$
$$= K_S^1(\omega, A_1) \int \mathbb{P}^2(d\omega_2')\mathbb{P}^2(A_2)$$
$$= \mathbb{P}^2(A_2)K_S^1(\omega, A_1).$$

*where we used the condition on $K_\emptyset$ in Definition 2.1. By the usual monotone convergence theorem arguments, we have that, for any $A \in \mathcal{H}^1 \otimes \mathcal{H}^2$,*

$$\int K_S^1(\omega, d\omega')\kappa(\omega', A)$$
$$= \int \kappa(\omega, d\omega_1' d\omega_2')K_S^1 \otimes K_\emptyset^2((\omega_1', \omega_2'), A_1 \times A_2).$$

*Thus, interventional consistency holds, in this case even for all sets $A$, not just for those measurable with respect to $\mathcal{H}^2_{\rho(T^1)}$.*

This shows that we can consider causal maps including our system into a larger system containing additional independent components.

Finally, we consider a more involved embedding example.

**Example 4.5** (Inclusion of SCMs). *Consider the following SCM*

$$H = N_H, \qquad X = H + N_X,$$

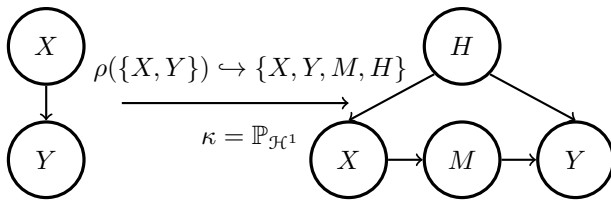

Figure 5: Inclusions of SCMs (Example 4.5).

$$M = X + N_M, \qquad Y = M + H + N_Y.$$

*We denote the joint distribution of $(X, Y, M, H)$ by $\mathbb{P}$, and the marginal distribution on $(X, Y)$ by $\mathbb{P}^{XY}$.*

*We consider a causal space $\mathcal{C}^1 = (\Omega^1, \mathcal{H}^1, \mathbb{P}^{XY}, \mathbb{K})$ that represents the pair $(X, Y)$, where $\Omega^1 = \mathbb{R}^2$ and $\mathcal{H}^1 = \mathcal{B}(\mathbb{R}^2)$, and a causal space $\mathcal{C}^2 = (\Omega^2, \mathcal{H}^2, \mathbb{P}, \mathbb{L})$ representing the full SCM, where $\Omega^2 = \mathbb{R}^4$ and $\mathcal{H}^2 = \mathcal{B}(\mathbb{R}^4)$, i.e., it contains in addition a mediator and a confounder. The causal mechanisms $\mathbb{K}$ and $\mathbb{L}$ are derived from the SCM. Then we consider the obvious $\rho$ that embeds $\{X, Y\}$ into $\{X, Y, M, H\}$ and, for $A \in \mathcal{H}^2$,*

$$\kappa(\cdot, A) = \mathbb{P}_{\mathcal{H}^1}(A).$$

*Clearly, this pair is admissible because on the variables $X$ and $Y$ we use the identity transformation. Distributional consistency follows by*

$$\int \kappa((x, y), A) \mathbb{P}^{XY}(d(x, y)) = \int \mathbb{P}_{\mathcal{H}^1}(A) d\mathbb{P}^{XY}$$
$$= \mathbb{P}(A).$$

*Interventional consistency also holds so that $(\kappa, \rho)$ is indeed a causal transformation. For a proof of this fact we refer to the more general result in Lemma 6.3.*

This example therefore shows that we can embed a system in a larger system that captures a more accurate description.

### 4.2 ABSTRACTIONS

Note that Example 4.3 is different from Examples 4.4 and 4.5 in that it compresses the representation while the other two consider an extension of the system. As these are different objectives, we consider the following definition.

**Definition 4.6** (Abstractions). The pair of maps $(\kappa, \rho)$ between measurable spaces $(\Omega^1, \mathcal{H}^1) = \otimes_{t \in T^1}(E_t, \mathcal{E}_t)$ and $(\Omega^2, \mathcal{H}^2) = \otimes_{t \in T^2}(E_t, \mathcal{E}_t)$ is called an *abstraction* if $\rho : T^1 \to T^2$ is surjective.

In the case of abstractions it is often sufficient to consider deterministic maps, motivating the following definition.

**Definition 4.7** (Perfect Abstractions). An abstraction $(\kappa, \rho)$ is called a perfect abstraction if $\kappa$ is deterministic, i.e., $\kappa = \kappa_f$ for some measurable $f : \Omega^1 \to \Omega^2$, and moreover $f$ is surjective.

We finally remark that one further setting of potential interest would be to consider the inverse of an abstraction, i.e., a setting where a summary variable $X$ is mapped to a more detailed description $(X_1, X_2)$. However, to accommodate such transformations we need a slightly different framework than the one presented here. Roughly, we need to consider $\rho : T^1 \to \mathcal{P}(T^2)$ with $\rho(t_1) \cap \rho(t_1') = \emptyset$ for $t_1, t_1' \in T_1$, and interventions on all sets $S \subset T^1$ can be expressed as interventions on the target $\mathcal{C}^2$ (i.e., the more fine-grained representations), while this is reversed in our case so that those two settings are dual to each other. We do not pursue this here any further, as those transformations are of more limited interest and applicability. Let us emphasise nevertheless that it seems ambitious to handle all cases in one framework. Indeed, combining variables in a summary variable or splitting variables in a more fine-grained description are meaningful operations, but it is less clear to interpret in a causal manner a definition of a transformation $(X_1, X_2) \to (Y_1, Y_2)$ that allows both at the same time. For example, intervening on $X_1$, in general, then does not correspond to a meaningful causal operation on the variables $(Y_1, Y_2)$. We also remark that this attempt has not been made in the SCM literature, where the focus is almost exclusively on abstractions.

## 5 COMPARISON WITH ABSTRACTION IN THE SCM FRAMEWORK

Rubenstein et al. [2017] gives the definition of *exact transformations* between SCMs. While being the seminal work on the theory of causal abstractions, it is probably also the most relevant to compare to our proposals. We first recall some essential aspects of their definition of SCMs (or SEMs, for structural equation models, by their nomenclature)[3].

**Definition 5.1** ([Rubenstein et al., 2017, Definition 1]). Let $\mathbb{I}_X$ be an index set. An SEM $\mathcal{M}_X$ over variables $X = (X_i : i \in \mathbb{I}_X$ taking values in $\mathcal{X}$ is a tuple $(\mathcal{S}_X, \mathbb{P}_E)$, where

- $\mathcal{S}_X$ is a set of structural equations, i.e. the set of equations $X_i = f_i(X, E_i)$ for $i \in \mathbb{I}_X$;
- $\mathbb{P}_E$ is a distribution over the exogenous variables $E = (E_i : i \in \mathbb{I}_X)$.

Note that their definition of SCMs is a bit more general than standard ones in the literature (e.g. [Peters et al., 2017,

---

[3]In this section, some imported notations might clash with ours; the clashes are restricted to this section and should not cause any confusion.

p.83, Definition 6.2]), in that they allow, for example, cycles and latent confounders, but they simply insist that there must be a unique solution to any interventions. They also consider a specific set of "allowed interventions", rather than considering all possible interventions. We also recall some essential aspects of the notion of exact transformations.

**Definition 5.2** ([Rubenstein et al., 2017, Definition 3]). Let $\mathcal{M}_X$ and $\mathcal{M}_Y$ be SCMs, and $\tau : \mathcal{X} \to \mathcal{Y}$ a function. We say that $\mathcal{M}_Y$ is an *exact $\tau$-transformation* of $\mathcal{M}_X$ if, there exists a surjective mapping $\omega$ of the interventions such that for any intervention $i$, $\mathbb{P}^i_{\tau(X)} = \mathbb{P}^{\omega(i)}_Y$.

Note that this definition is trying to capture the same concept as our notion of interventional consistency given in (2): that interventions and transformations commute. However, there are several aspects in which our proposal is more appealing.

- They only consider deterministic maps $\tau : \mathcal{X} \to \mathcal{Y}$, whereas we allow the map $\rho$ to be stochastic.
- They have to find a separate map $\omega$ *between the interventions themselves*, whereas our map $\rho$ also determines the transformation of the causal kernels.
- By insisting on surjectivity of $\omega$, they only allow the consideration of abstraction, whereas we can consider more general transformations of causal spaces, such as inclusions considered in Example 4.4.

Nevertheless, restricted to considerations amenable to both approaches, the notions coincide. For example, we return to Example 4.3, where we already showed that $f_*\mathbb{P} = \mathbb{Q}$, $f_*K_{\{X_1,X_2\}} = L_X$ and $f_*K_{\{Y_1,Y_2\}} = L_Y$, which implies that two-variable SCM is an exact transformation of the four-variable SCM according to Definition 5.2.

Finally, we mention that Beckers and Halpern [2019] criticise exact transformations of Rubenstein et al. [2017] on the basis that probabilities and allowed interventions can mask significant differences between SCMs, and then proceed to propose definitions of abstractions that depend only on the structural equations, independently of probabilities. We remark that this criticism is not valid in our framework, in that the interventional consistency of our transformations is imposed independently of probabilities, making it impossible to mask them with the choice of probability measures. That this is possible with SCMs is an artifact of the fact that in SCMs, the observational and interventional measures are coupled through the exogenous distribution, whereas in causal spaces they are completely decoupled. Moreover, we consider all possible interventions rather than a reduced set of allowed interventions. We also remark that, since probabilities and causal kernels are the primitive objects in our framework, rather than being derived by other primitive objects (namely the structural equations), it does not make sense for the transformation to be defined independently of probabilities, as done by Beckers and Halpern [2019].

# 6 FURTHER PROPERTIES OF CAUSAL TRANSFORMATIONS

In this section we investigate various properties of causal transformations and connect them to the notions introduced in Section 2.

## 6.1 EXISTENCE AND UNIQUENESS OF CAUSAL TRANSFORMATIONS

First, we have the following lemma on the composition of causal transformations. Recall that for two probability kernels $\kappa_1 : \Omega^1 \times \mathcal{H}^2 \to [0,1]$ mapping $(\Omega^1, \mathcal{H}^1)$ to $(\Omega^2, \mathcal{H}^2)$ and $\kappa_2 : \Omega^2 \times \mathcal{H}^3 \to [0,1]$ mapping $(\Omega^2, \mathcal{H}^2)$ to $(\Omega^3, \mathcal{H}^3)$ the concatenation defined by [Cinlar, 2011, p.39]

$$\kappa_1 \circ \kappa_2(\omega_1, A) = \int \kappa_1(\omega_1, d\omega_2)\kappa_2(\omega_2, A)$$

defines a probability kernel from $(\Omega^1, \mathcal{H}^1)$ to $(\Omega^3, \mathcal{H}^3)$.

**Lemma 6.1** (Compositions of Causal Transformations). *Let $(\kappa_1, \rho_1) : \mathcal{C}^1 \to \mathcal{C}^2$ and $(\kappa_2, \rho_2) : \mathcal{C}^2 \to \mathcal{C}^3$ be causal transformations. If $(\kappa_1, \rho_1)$ is an abstraction then $(\kappa_3, \rho_3) = (\kappa_1 \circ \kappa_2, \rho_1 \circ \rho_2) : \mathcal{C}^1 \to \mathcal{C}^3$ is a causal transformation.*

The proof can be found in Appendix B.2. We remark that, unfortunately, we cannot remove the assumption that the first transformation is an abstraction. Let us clarify this through an example.

**Example 6.2.** *Consider an SCM with equations*

$$\begin{aligned} X_1 &= N_1, \\ X_2 &= N_2, \\ Y &= X_1 + X_2 + N_Y \end{aligned}$$

*where $N_1$, $N_2$, and $N_Y$ follow independent standard normal distributions. Then we can consider the causal space $\mathcal{C}^1$ containing $(X_1, Y)$, the causal space $\mathcal{C}^2$ containing $(X_1, X_2, Y)$ and an abstraction $\mathcal{C}^3$ containing $(X_1 + X_2, Y)$. Then we can embed $\mathcal{C}^1 \to \mathcal{C}^2$ and there is an abstraction $\mathcal{C}^2 \to \mathcal{C}^3$ which are both transformations of causal spaces.*

*However, their concatenation is not a causal transformation because it is not even admissible (and also interventional consistency does not hold for the intervention $K_X^3$ as this cannot be expressed by $K_{X_1}^1$). Note that $P_{X_2|X_1=x_1,Y=y} = N((y-x_1)/2, 1/2)$ and therefore we have $\kappa_1((x_1, y), \cdot) = \delta_{x_1} \otimes N((y-x_1)/2, 1/2) \otimes \delta_y$.*

*We also have $\kappa_2((x_1, x_2, y), \cdot) = \delta_{(x_1+x_2, y)}$. Thus, their concatenation is given by*

$$\kappa_3((x_1, y), \cdot) = N((y+x_1)/2, 1/2) \otimes \delta_y.$$

*So the first coordinate is not measurable with respect to $\mathcal{H}^1_{X_1}$.*

This shows that we lose measurability along the concatenation because the variables added in the more complete description $\mathcal{C}^2$ may depend on all other variables.

Let us now generalise Example 4.5 to general SCMs.

**Lemma 6.3** (Inclusion of SCMs). *Consider an acyclic SCM on endogenous variables $(X_1, \ldots, X_d) \in \mathbb{R}^d$ with observational distribution $\mathbb{P}$. Let $S \subset [d]$, $R = S^c = [d] \setminus S$ and consider causal spaces $\mathcal{C}^1 = (\Omega^1, \mathcal{H}^1, \mathbb{P}^S, \mathbb{K})$ and $\mathcal{C}^2 = (\Omega^2, \mathcal{H}^2, \mathbb{P}, \mathbb{L})$, where we have $(\Omega^1, \mathcal{H}^1) = (\mathbb{R}^{|S|}, \mathcal{B}(\mathbb{R}^{|S|}))$ and $(\Omega^2, \mathcal{H}^2) = (\mathbb{R}^d, \mathcal{B}(\mathbb{R}^d))$. Moreover, $\mathbb{P}^S$ is the marginal distribution on the variables in $S$, and the causal mechanisms $\mathbb{K}$ and $\mathbb{L}$ are derived from the SCM. In particular, $\mathbb{K}$ is a marginalisation of $\mathbb{L}$, namely, for any $\omega \in \Omega^2$, any event $A \in \mathcal{H}^1$ and any $S' \subseteq S$, we have that $K_{S'}(\omega, A) = L_{S'}(\omega, A)$.*

*Consider the map $\rho : S \hookrightarrow [d]$ and $\kappa(\cdot, A) = \mathbb{P}_{\mathcal{H}^1}(A)$. Then $(\rho, \kappa)$ is a causal transformation from $\mathcal{C}^1$ to $\mathcal{C}^2$.*

The proof can be found in Appendix B.2.

We now investigate to what degree distributional and interventional consistency determines the causal structure on the target space. We show that generally the causal structure on $\mathcal{H}^2_{\rho(T)}$ is quite rigid.

**Lemma 6.4** (Rigidity of target causal structure). *Let $\mathcal{C}^2 = (\Omega^2, \mathcal{H}^2, \mathbb{P}^2, \mathbb{K}^2)$ and $\tilde{\mathcal{C}}^2 = (\Omega^2, \mathcal{H}^2, \tilde{\mathbb{P}}^2, \tilde{\mathbb{K}}^2)$ be two causal spaces with the same underlying measurable space. Let $(\kappa, \rho)$ be an admissible pair for the measurable spaces $(\Omega^1, \mathcal{H}^1)$ and $(\Omega^2, \mathcal{H}^2)$. Assume that the pair $(\kappa, \rho)$ defines causal transformations $\varphi : \mathcal{C}^1 \to \mathcal{C}^2$ and $\tilde{\varphi} : \mathcal{C}^1 \to \tilde{\mathcal{C}}^2$ be a causal transformations. Then $\mathbb{P}^2 = \tilde{\mathbb{P}}^2$, and for all $A \in \mathcal{H}^2_{\rho(T^1)}$ and any $S \subset T^2$*

$$K_S^2(\omega, A) = \tilde{K}_S^2(\omega, A) \quad \text{for } \mathbb{P}^2 = \tilde{\mathbb{P}}^2 \text{ a. e. } \omega \in \Omega^2.$$

The proof is in Appendix B.2. We cannot expect to derive much stronger results for general causal transformation because interventional consistency does not restrict $K^2(\omega, A)$ for $\omega$ not in the support of $\mathbb{P}^2$ or $A \notin \mathcal{H}^2_{\rho(T)}$. E.g., in the setting of Example 4.4 the causal structure on the second factor is arbitrary.

However, when we consider deterministic maps $(f, \rho)$ such that $f : (\Omega^1, \mathcal{H}^1) \to (\Omega^2, \mathcal{H}^2)$ and $\rho$ are surjective, then there is at most one causal structure on the target space $(\Omega^2, \mathcal{H}^2)$ such that the pair $(f, \rho)$ is a causal transformation (and thus a perfect abstraction).

**Lemma 6.5** (Surjective Deterministic Maps). *Suppose $(f, \rho)$ is an admissible pair for the causal space $\mathcal{C}^1 = (\Omega^1, \mathcal{H}^1, \mathbb{P}^1, \mathbb{K}^1)$ to the measurable space $X^2 = (\Omega^2, \mathcal{H}^2)$ and assume that $\rho$ is surjective and $f : \Omega_1 \to \Omega_2$ measurable. If $f$ is surjective, there exists at most one causal space $\mathcal{C}^2 = (\Omega^2, \mathcal{H}^2, \mathbb{P}^2, \mathbb{K}^2)$ such that $(f, \rho) : \mathcal{C}^1 \to \mathcal{C}^2$ is a causal transformation.*

*If, in addition, $K^1_{\rho^{-1}(S^2)}(\cdot, A)$ is measurable with respect to $f^{-1}(\mathcal{H}^2_{S^2})$ for all $A \in f^{-1}(\mathcal{H}^2)$ and all $S^2 \subset T^2$ then a unique causal space $\mathcal{C}^2$ exists such that $(f, \rho) : \mathcal{C}^1 \to \mathcal{C}^2$ is a causal transformation.*

The proof is in Appendix B.2. To motivate the measurability condition for $K^1(\cdot, A)$ we remark that interventional consistency requires $K^1(\omega, A) = K^1(\omega', A)$ for $\omega, \omega'$ with $f(\omega) = f(\omega')$ and the measurability condition in the result is a slightly stronger condition than this.

Next, we show that interventions on a space can be pushed forward along a perfect abstraction.

**Lemma 6.6** (Perfect Abstraction on Intervened Spaces). *Let $\mathcal{C}^1 = (\Omega^1, \mathcal{H}^1, \mathbb{P}^1, \mathbb{K}^1)$ with $(\Omega^1, \mathcal{H}^1)$ a product with index set $T^1$ and $\mathcal{C}^2 = (\Omega^2, \mathcal{H}^2, \mathbb{P}^2, \mathbb{K}^2)$ with $(\Omega^2, \mathcal{H}^2)$ a product with index set $T^2$ be causal spaces, and let $(f, \rho) : \mathcal{C}^1 \to \mathcal{C}^2$ be a perfect abstraction.*

*Let $U^1 = \rho^{-1}(U^2) \subseteq T^1$ for some $U^2 \subseteq T^2$. Let $\mathbb{Q}^1$ be a probability measure on $(\Omega^1, \mathcal{H}^1_{U^1})$ and $\mathbb{L}^1$ a causal mechanism on $(\Omega^1, \mathcal{H}^1_{U^1}, \mathbb{Q}^1)$. Suppose that, for all $S \subseteq U^2$ and $A \in \mathcal{H}^1$, the map $L^1_{\rho^{-1}(S)}(\cdot, A)$ is measurable with respect to $f^{-1}(\mathcal{H}^2_S)$, and consider the intervened causal spaces*

$$\mathcal{C}^1_I = (\Omega^1, \mathcal{H}^1, (\mathbb{P}^1)^{\mathrm{do}(U^1, \mathbb{Q}^1)}, (\mathbb{K}^1)^{\mathrm{do}(U^1, \mathbb{Q}^1, \mathbb{L}^1)}),$$
$$\mathcal{C}^2_I = (\Omega^2, \mathcal{H}^2, (\mathbb{P}^2)^{\mathrm{do}(U^2, \mathbb{Q}^2)}, (\mathbb{K}^2)^{\mathrm{do}(U^2, \mathbb{Q}^2, \mathbb{L}^2)}),$$

*where $\mathbb{Q}^2 = f_* \mathbb{Q}^1$ and $\mathbb{L}^2$ is the unique family of kernels satisfying $L^2_S(f(\omega), A) = L^1_{\rho^{-1}(S)}(\omega, f^{-1}(A))$ for all $\omega \in \Omega^1$, $A \in \mathcal{H}^2$, and $S \subseteq U^2$. Then $(f, \rho) : \mathcal{C}^1_I \to \mathcal{C}^2_I$ is a perfect abstraction.*

The proof of this result is in Appendix B.2.

## 6.2 SOURCES AND CAUSAL EFFECTS UNDER CAUSAL TRANSFORMATIONS

We now study whether causal effects in target and domain of a causal transformation can be related. Our first results shows that for perfect abstractions having no causal effect in the domain implies that there is also no causal effect in the target.

**Lemma 6.7** (Perfect Abstraction and No Causal Effect). *Let $(f, \rho) : \mathcal{C}^1 \to \mathcal{C}^2$ be a perfect abstraction. Consider two sets $U^2, V^2 \subset T^2$ and denote $U^1 = \rho^{-1}(U^2)$ and $V^1 = \rho^{-1}(V^2)$. If $\mathcal{H}^1_{U^1}$ has no causal effect on $\mathcal{H}^1_{V^1}$ in $\mathcal{C}^1$, then $\mathcal{H}^2_{U^2}$ has no causal effect on $\mathcal{H}^2_{V^2}$ in $\mathcal{C}^2$.*

The proof can be found in Appendix B.3.

On the other hand, we can show that when there is an active causal effect in the target space, there is also an active causal effect in the domain.

**Lemma 6.8** (Perfect Abstraction and Active Causal Effects)**.** *Let* $(f, \rho) : \mathbb{C}^1 \to \mathbb{C}^2$ *be a perfect abstraction. Consider two sets* $U^2, V^2 \subset T^2$ *and denote* $U^1 = \rho^{-1}(U^2)$ *and* $V^1 = \rho^{-1}(V^2)$. *Assume that* $\mathcal{H}^2_{U^2}$ *has an active causal effect on* $\mathcal{H}^2_{V^2}$ *in* $\mathbb{C}^2$. *Then* $\mathcal{H}^1_{U^1}$ *has an active causal effect on* $\mathcal{H}^1_{V^1}$ *in* $\mathbb{C}^1$.

The proof is in Appendix B.3.

The reverse statements are not true, i.e., if there is no causal effect in the target there might be a causal effect in the domain and if there is an active causal effect in the domain this does not imply that ther is a causal effect in the target, which can be seen by considering a target space with only a single point. We can also study causal effects in the context of embedding transformations, as in Lemma 6.3. Then we see directly that active causal effects are preserved. On the other hand, it is straightforward to construct examples where there is no causal effect in a subsystem, but there is a causal effect in a larger system. This can be achieved by a violation of faithfulness.

**Example 6.9.** *Consider the SCM*

$$X = N_X,$$
$$M = N_X + N_M,$$
$$Y = M - X + N_Y.$$

*Then there is no causal effect from* $\sigma(X)$ *to* $\sigma(Y)$ *in the system* $(X, Y)$ *but there is a causal effect in the complete system.*

Finally, we show that similar results can be established for sources. Indeed, perfect abstraction preserve sources in the following sense.

**Lemma 6.10** (Perfect Abstraction and Sources)**.** *Let* $(f, \rho) : \mathbb{C}^1 \to \mathbb{C}^2$ *be a perfect abstraction. Consider two sets* $U^2, V^2 \subset T^2$ *and denote* $U^1 = \rho^{-1}(U^2)$ *and* $V^1 = \rho^{-1}(V^2)$. *Assume that* $\mathcal{H}^1_{U^1}$ *is a local source of* $\mathcal{H}^1_{V^1}$ *in* $\mathbb{C}^1$. *Then* $\mathcal{H}^2_{U^2}$ *is a local source of* $\mathcal{H}^2_{V^2}$ *in* $\mathbb{C}^2$.

*In particular, this implies that if* $\mathcal{H}^1_{U^1}$ *is a global source then* $\mathcal{H}^2_{U^2}$ *also is a global source.*

The proof is in Appendix B.3.

Similar to our results for causal effects, the existence of sources in the abstracted space does not guarantee the existence of sources in the domain space. Note that local sources are preserved in the setting of Lemma 6.3. On the other hand, global sources are clearly not preserved, as we can add a global source to the system.

## 7 CONCLUSION

In this paper, we developed the theory of causal spaces initiated by Park et al. [2023] by proposing the notions of

products of causal spaces and transformations of causal spaces. They are defined via natural extensions of the notions of products and probability kernels in probability theory. Not only are they mathematically elegant objects, but they have natural and important semantic interpretations as causal independence and abstraction or inclusion of causal spaces. Moreover, we explore the connections of these notions to those of causal effects and sources introduced in [Park et al., 2023].

Despite the beauty and practical usefulness of the structural causal model and potential outcomes frameworks, we believe that the theory of causal spaces has the potential to overcome some of the longstanding limitations of them in terms of rigour and expressiveness, and the contribution of this paper is to develop the theory further in terms of treating multiple causal spaces through products and transformations rather than focusing the investigation to single causal spaces.

Although probability theory does not seem to be so amenable to a category-theoretic treatment as other mathematical objects, there have been some efforts to do so [Lynn, 2010, Adachi and Ryu, 2016, Cho and Jacobs, 2019, Fritz, 2020]. As future work, it would be interesting to explore extensions of the transformations proposed here to formal category-theoretic morphisms between causal spaces.

### Acknowledgements

This work was supported by the Tübingen AI Center.

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

# Products, Abstractions, and Inclusions of Causal Spaces
## (Supplementary Material)

**Simon Buchholz**[*1]  **Junhyung Park**[*1]  **Bernhard Schölkopf**[1]

[1]Empirical Inference Department, Max Planck Institute for Intelligent Systems, Tübingen, Germany

In the supplementary material, we provide the missing proofs for the results in our paper, and we recall some definitions from probability theory.

## A    BACKGROUND ON PROBABILITY THEORY

Let us collect some definitions and notations from probability theory. Recall that a probability space $(\Omega, \mathcal{H}, \mathbb{P})$ is defined through the following axioms:

(i) $\Omega$ is a set;

(ii) $\mathcal{H}$ is a collection of subsets of $\Omega$, called *events*, such that

- $\Omega \in \mathcal{H}$;
- if $A \in \mathcal{H}$, then $\Omega \setminus A \in \mathcal{H}$;
- if $A_1, A_2, ... \in \mathcal{H}$, then $\cup_n A_n \in \mathcal{H}$;

(iii) $\mathbb{P}$ is a probability measure on $(\Omega, \mathcal{H})$, i.e. a function $\mathbb{P} : \mathcal{H} \to [0, 1]$ satisfying

- $\mathbb{P}(\emptyset) = 0$;
- $\mathbb{P}(\cup_n A_n) = \sum_n \mathbb{P}(A_n)$ for any disjoint sequence $A_n$ in $\mathcal{H}$;
- $\mathbb{P}(\Omega) = 1$.

Probability kernels (sometimes called Markov kernels) from $(\Omega^1, \mathcal{H}^1)$ to $(\Omega^2, \mathcal{H}^2)$ are maps $\kappa : \Omega^1 \times \mathcal{H}^2 \to [0, 1]$ such that

(i) For every $A \in \mathcal{H}^2$ the function

$$\omega \to \kappa(\omega, A)$$

is measurable with respect to $\mathcal{H}^1$.

(ii) For every $\omega \in \Omega^1$ the function

$$A \to \kappa(\omega, A)$$

defines a measure on $(\Omega^2, \mathcal{H}^2)$.

See, for example, [Cinlar, 2011, p.37, Section I.6] for details.

We remark that we can concatenate probability kernels and we can consider product kernels.

We denote integration with respect to a (probability) measure $\mathbb{P}$ by

$$\int \mathbb{P}(\mathrm{d}\omega) T(\omega)$$

---

[*]Equal Contribution

for a measurable map $T : (\Omega, \mathcal{H}) \to (\mathbb{R}, \mathcal{B}(\mathbb{R}))$.

For a measurable map $f : (\Omega^1, \mathcal{H}^1) \to (\Omega^2, \mathcal{H}^2)$ we define the pushforward measure $f_*\mathbb{P}$ by $f_*\mathbb{P}(A) = \mathbb{P}(f^{-1}(A))$. The transformation formula for the pushforward measure reads

$$\int f_*\mathbb{P}(\mathrm{d}\omega)T(\omega) = \int \mathbb{P}(\mathrm{d}\omega')T \circ f(\omega').$$

Finally we recall the Factorisation lemma [Cinlar, 2011, p.76, Theorem II.4.4].

**Lemma A.1** (Factorisation Lemma). *Let $T : \Omega^1 \to (\Omega^2, \mathcal{H}^2)$ be a function. A function $f : \Omega^1 \to \mathbb{R}$ is $\sigma(T)$-$\mathcal{B}(\mathbb{R})$ measurable if and only if there is a measurable function $g : (\Omega^2, \mathcal{H}^2) \to (\mathbb{R}, \mathcal{B}(\mathbb{R}))$ such that $f = g \circ T$.*

# B    PROOFS

In this appendix we collect the proofs of all the results in the paper.

## B.1    PROOFS FOR SECTION 3

**Lemma 3.2** (Products of Causal Spaces are Causal Spaces). *The product causal space $\mathcal{C}^1 \otimes \mathcal{C}^2$ as defined in Definition 3.1 is a causal space.*

*Proof of Lemma 3.2.*  It is a standard fact that $\mathbb{K}^1 \otimes \mathbb{K}^2$ defines a family of probability kernels[1]. For the first axiom of causal kernels (Definition 2.1(i)), we observe that

$$
\begin{aligned}
(K^1 \otimes K^2)_\emptyset((\omega_1, \omega_2), A_1 \times A_2) &= K^1_\emptyset(\omega_1, A_1)K^2_\emptyset(\omega_2, A_2) \\
&= \mathbb{P}^1(A_1)\mathbb{P}^2(A_2) \\
&= \mathbb{P}^1 \otimes \mathbb{P}^2(A_1 \times A_2).
\end{aligned}
$$

By standard reasoning based on the monotone class theorem, this extends to $A \in \mathcal{H}^1 \otimes \mathcal{H}^2$ and therefore the first axiom of causal spaces is satisfied. For the second axiom of causal spaces, for any $S = S^1 \cup S^2$, first fix arbitrary $A_1 \in \mathcal{H}^1_{S^1}$ and $A_2 \in \mathcal{H}^1_{S^2}$. Then, for all $B_1 \in \mathcal{H}^1$ and $B_2 \in \mathcal{H}^2$, we find that for all $\omega = (\omega_1, \omega_2)$,

$$
\begin{aligned}
L_S(\omega, (A_1 \times A_2) \cap (B_1 \times B_2)) &= K^1_{S_1}(\omega_1, A_1 \cap B_1)K^2_{S^2}(\omega_2, A_2 \cap B_2) \\
&= \mathbf{1}_{A_1}(\omega_1)K^1_{S^1}(\omega_1, B_1)\mathbf{1}_{A_2}(\omega_2)K^2_{S^2}(\omega_2, B_2) \\
&= \mathbf{1}_{A_1 \times A_2}(\omega)L_S(\omega, B_1 \times B_2).
\end{aligned}
$$

Hence, for this fixed pair $A_1$, $A_2$ and this $\omega$, the measures $B \mapsto L_S(\omega, (A_1 \times A_2) \cap B)$ and $B \mapsto \mathbf{1}_{A_1 \times A_2}(\omega)L_S(\omega, B)$ are identical on the generating rectangles $B_1 \times B_2$, hence they are identical on all of $\mathcal{H}^1 \otimes \mathcal{H}^2$ by the standard monotone class theorem reasoning. Now, since this is true for arbitrary rectangles $A_1 \times A_2$ with $A_1 \in \mathcal{H}^1_{S^1}$ and $A_2 \in \mathcal{H}^2_{S^2}$, if we now fix $B \in \mathcal{H}^1 \otimes \mathcal{H}^2$, we have that the two measures $A \mapsto L_S(\omega, A \cap B)$ and $A \mapsto \mathbf{1}_A(\omega)L_S(\omega, B)$ on $\mathcal{H}^1_{S^1} \otimes \mathcal{H}^2_{S^2}$ are identical on the generating rectangles $A_1 \times A_2$, hence they are identical on all of $\mathcal{H}^1_{S^1} \otimes \mathcal{H}^2_{S^2}$. Now both $A$ and $B$ are arbitrary elements of $\mathcal{H}^1 \otimes \mathcal{H}^2$ and $\mathcal{H}^1_{S^1} \otimes \mathcal{H}^2_{S^2}$ respectively. To conclude, we have, for all $\omega$, $A \in \mathcal{H}^1 \otimes \mathcal{H}^2$ and $B \in \mathcal{H}^1_{S^1} \otimes \mathcal{H}^2_{S^2}$,

$$L_S(\omega, A \cap B) = \mathbf{1}_A(\omega)L_S(\omega, B),$$

confirming the second axiom of causal spaces.    $\square$

**Lemma 3.3** (Causal Effects in Product Spaces). *Suppose $\mathcal{C}^1 = (\Omega^1, \mathcal{H}^1, \mathbb{P}^1, \mathbb{K}^1)$ and $\mathcal{C}^2 = (\Omega^2, \mathcal{H}^2, \mathbb{P}^2, \mathbb{K}^2)$ with $\Omega^1 = \times_{t \in T^1} E_t$ and $\Omega^2 = \times_{t \in T^2} E_t$ are two causal spaces. Then in $\mathcal{C}^1 \otimes \mathcal{C}^2$,*

 *(i)  $\mathcal{H}_{T^1}$ has no causal effect on $\mathcal{H}_{T^2}$, and $\mathcal{H}_{T^2}$ has no causal effect on $\mathcal{H}_{T^1}$;*

 *(ii)  $\mathcal{H}_{T^1}$ and $\mathcal{H}_{T^2}$ are (local) sources of each other.*

---

[1]See, e.g. math.stackexchange.com/questions/84078/product-of-two-probability-kernel-is-a-probability-kernel

*Proof of Lemma 3.3.* (i) Denote the causal kernels on the product space by $K^p$. Take any event $A \in \mathcal{H}_{T^2}$, and any $S \subseteq T^1 \cup T^2$. Note that $S$ can be written as a union $S = S^1 \cup S^2$ for some $S^1 \subseteq T^1$ and $S^2 \subseteq T^2$. Then see that, by writing $A = \Omega^1 \times A' \in \mathcal{H}_{T^1} \otimes \mathcal{H}_{T^2}$ with $A' \subseteq \Omega^2$,

$$
\begin{aligned}
K_S^p(\omega, A) &= K_{S^1 \cup S^2}^p(\omega, A) \\
&= K_{S^1}^1(\omega, \Omega_1) K_{S^2}^2(\omega, A') \\
&= K_\emptyset^1(\omega, \Omega_1) K_{S^2}(\omega, A') \\
&= K_{S \setminus T^1}(\omega, A).
\end{aligned}
$$

Here we used that $K_{S^1}^1(\omega, \Omega_1) = 1 = K_\emptyset^1(\omega, \Omega_1)$ because $K(\omega, \cdot)$ is a probability measure for a probability kernel. So $\mathcal{H}_{T^1}$ has no causal effect on $A$. Implication in the other direction follows the same argument.

(ii) Take any $A \in \mathcal{H}_{T^2}$. By (i), $\mathcal{H}_{T^1}$ has no causal effect on $A$, so

$$
K_{T^1}(\omega, A) = K_{T^1 \setminus T^1}(\omega, A) = K_\emptyset(\omega, A) = \mathbb{P}(A).
$$

But since $\mathcal{H}_{T^1}$ and $\mathcal{H}_{T^2}$ are probabilistically independent, $\mathbb{P}_{T^1}(A) = \mathbb{P}(A)$. Hence, $\mathbb{P}_{T^1}(A) = K_{T^1}(\omega, A)$, meaning $\mathcal{H}_{T_1}$ is a source of $A$. Since $A \in \mathcal{H}_{T_2}$ was arbitrary, $\mathcal{H}_{T_1}$ is a source of $\mathcal{H}_{T_2}$. The implication in the other direction follows the same argument.

$\square$

## B.2 PROOFS FOR SECTION 6.1

**Lemma 6.1** (Compositions of Causal Transformations). *Let $(\kappa_1, \rho_1) : \mathcal{C}^1 \to \mathcal{C}^2$ and $(\kappa_2, \rho_2) : \mathcal{C}^2 \to \mathcal{C}^3$ be causal transformations. If $(\kappa_1, \rho_1)$ is an abstraction then $(\kappa_3, \rho_3) = (\kappa_1 \circ \kappa_2, \rho_1 \circ \rho_2) : \mathcal{C}^1 \to \mathcal{C}^3$ is a causal transformation.*

*Proof of Lemma 6.1.* First, we claim that the pair $(\kappa_3, \rho_3) = (\kappa_1 \circ \kappa_2, \rho_1 \circ \rho_2)$ is admissible. We have to show that, for any $S^3 \subset \rho_3(T^1)$ and $A \in \mathcal{H}_{S^3}^3$, the map $\kappa_3(\cdot, A)$ is measurable with respect to $\mathcal{H}_{\rho_3^{-1}(S^3)}^1$.

Let us call $\rho_2^{-1}(S^3) = S^2$. Note that, since $(\kappa_2, \rho_2) : \mathcal{C}^2 \to \mathcal{C}^3$ is a causal transformation, $\kappa_2(\cdot, A)$ is measurable with respect to $\mathcal{H}_{S^2}^2$. Since we assume that the first map is an abstraction, we find that $S^2 \subset \rho^1(T^1) = T^2$, and thus by Definition 4.1 that for $B \in \mathcal{H}_{S^2}^2$ the function $\kappa_1(\cdot, B)$ is measurable with respect to $\mathcal{H}_{\rho_3^{-1}(S^3)}^1$, where we used $\rho_3^{-1}(S^3) = \rho_1^{-1}(S^2)$. We now use the relation $\kappa_3(\omega, A) = \int \kappa_1(\omega, d\omega') \kappa_2(\omega', A)$. Since $\kappa_2(\cdot, A)$ is measurable with respect to $\mathcal{H}_{S^2}^2$, we conclude that we can approximate $\kappa_2(\cdot, A)$ by a simple function $\sum \alpha_i \mathbf{1}_{B_i}(\cdot)$ with $B_i \in \mathcal{H}_{S^2}^2$. But for such a simple function, we find

$$
\int \kappa_1(\omega, d\omega') \sum_i \alpha_i \mathbf{1}_{B_i}(\omega') = \sum_i \alpha_i \kappa_1(\omega, B_i),
$$

which is measurable with respect to $\mathcal{H}_{S^1}^1$ as a sum of measurable functions because $(\kappa_1, \rho_1)$ is admissible. By passing to the limit $(\kappa_3, \rho_3)$ is admissible.

Next we show that distributional consistency holds, which follows directly from distributional consistency of $(\kappa_1, \rho_1)$ and $(\kappa_2, \rho_2)$:

$$
\begin{aligned}
\int \mathbb{P}^1(d\omega) \kappa_3(\omega, A) &= \int \mathbb{P}^1(d\omega) \kappa_1(\omega, d\omega_2) \kappa_2(\omega_2, A) \\
&= \int \mathbb{P}^2(d\omega_2) \kappa_2(\omega_2, A) \\
&= \mathbb{P}^3(A).
\end{aligned}
$$

Next we consider interventional consistency. Let $S^3 \subset \rho_3(T^1)$ and define $S^2 = \rho_2^{-1}(S^3)$ and $S^1 = \rho_3^{-1}(S^1) = \rho_1^{-1}(S^2)$. Note that, since $(\kappa_1, \rho_1)$ is an abstraction, i.e., $\rho_1$ is surjective, we have $S^2 \subset \rho_1(T^1) = T^2$. Now we find that, for $\omega_1 \in \Omega^1$ and $A \in \mathcal{H}^3$,

$$
\int \kappa_3(\omega, d\omega') K_{S_3}^3(\omega', A) = \int \kappa_1(\omega, d\omega_2) \kappa_2(\omega_2, d\omega') K_{S_3}^3(\omega', A)
$$

$$= \int \kappa_1(\omega, d\omega_2) K^2_{S_2}(\omega_2, d\omega') \kappa_2(\omega', A)$$

$$= \int K^1_{S_1}(\omega, d\omega') \kappa_1(\omega', d\omega_2) \kappa_2(\omega_2, A)$$

$$= \int K^1_{S_1}(\omega, d\omega') \kappa_3(\omega', A).$$

This ends the proof as we have shown that $(\kappa_3, \rho_3)$ is a causal transformation. $\qquad\square$

**Lemma 6.3** (Inclusion of SCMs). *Consider an acyclic SCM on endogenous variables $(X_1, \ldots, X_d) \in \mathbb{R}^d$ with observational distribution $\mathbb{P}$. Let $S \subset [d]$, $R = S^c = [d] \setminus S$ and consider causal spaces $\mathcal{C}^1 = (\Omega^1, \mathcal{H}^1, \mathbb{P}^S, \mathbb{K})$ and $\mathcal{C}^2 = (\Omega^2, \mathcal{H}^2, \mathbb{P}, \mathbb{L})$, where we have $(\Omega^1, \mathcal{H}^1) = (\mathbb{R}^{|S|}, \mathcal{B}(\mathbb{R}^{|S|}))$ and $(\Omega^2, \mathcal{H}^2) = (\mathbb{R}^d, \mathcal{B}(\mathbb{R}^d))$. Moreover, $\mathbb{P}^S$ is the marginal distribution on the variables in $S$, and the causal mechanisms $\mathbb{K}$ and $\mathbb{L}$ are derived from the SCM. In particular, $\mathbb{K}$ is a marginalisation of $\mathbb{L}$, namely, for any $\omega \in \Omega^2$, any event $A \in \mathcal{H}^1$ and any $S' \subseteq S$, we have that $K_{S'}(\omega, A) = L_{S'}(\omega, A)$.*

*Consider the map $\rho : S \hookrightarrow [d]$ and $\kappa(\cdot, A) = \mathbb{P}_{\mathcal{H}^1}(A)$. Then $(\rho, \kappa)$ is a causal transformation from $\mathcal{C}^1$ to $\mathcal{C}^2$.*

*Proof of Lemma 6.3.* First we note that as in Example 4.5 it is clear that $(\kappa, \rho)$ is admissible and

$$\int \kappa(x_S, A) \mathbb{P}^S(dx_S) = \int \mathbb{P}_{\mathcal{H}^1}(A) d\mathbb{P}^S = \mathbb{P}(A),$$

so we have distributional consistency.

For interventional consistency, let $A \in \mathcal{H}^1$, $S' \subseteq S$ and $\omega \in \Omega^1$ be arbitrary. Then see that

$$\int K_{S'}(\omega, d\omega') \kappa(\omega', A) = \int K_{S'}(\omega, d\omega') \mathbb{P}_{\mathcal{H}^1}(\omega', A)$$

$$= \int K_{S'}(\omega, d\omega') \mathbf{1}_A(\omega') \qquad \text{since } A \in \mathcal{H}^1$$

$$= K_{S'}(\omega, A).$$

On the other hand, see that

$$\int \kappa(\omega, d\omega') L_{S'}(\omega', A) = \int \mathbb{P}_{\mathcal{H}^1}(\omega, d\omega') L_{S'}(\omega', A)$$

$$= \int \mathbf{1}_{d\omega'}(\omega) L_{S'}(\omega', A) \qquad \text{since } L_{S'}(\cdot, A) \text{ is measurable with respect to } \mathcal{H}^1$$

$$= L_{S'}(\omega, A).$$

But by the marginalisation condition on the causal mechanisms $\mathbb{K}$ and $\mathbb{L}$, we have that $L_{S'}(\omega, A) = K_{S'}(\omega, A)$ for all $\omega \in \Omega^1$. This proves interventional consistency.

$\qquad\square$

**Lemma 6.4** (Rigidity of target causal structure). *Let $\mathcal{C}^2 = (\Omega^2, \mathcal{H}^2, \mathbb{P}^2, \mathbb{K}^2)$ and $\tilde{\mathcal{C}}^2 = (\Omega^2, \mathcal{H}^2, \tilde{\mathbb{P}}^2, \tilde{\mathbb{K}}^2)$ be two causal spaces with the same underlying measurable space. Let $(\kappa, \rho)$ be an admissible pair for the measurable spaces $(\Omega^1, \mathcal{H}^1)$ and $(\Omega^2, \mathcal{H}^2)$. Assume that the pair $(\kappa, \rho)$ defines causal transformations $\varphi : \mathcal{C}^1 \to \mathcal{C}^2$ and $\tilde{\varphi} : \mathcal{C}^1 \to \tilde{\mathcal{C}}^2$ be a causal transformations. Then $\mathbb{P}^2 = \tilde{\mathbb{P}}^2$, and for all $A \in \mathcal{H}^2_{\rho(T^1)}$ and any $S \subset T^2$*

$$K^2_S(\omega, A) = \tilde{K}^2_S(\omega, A) \quad \text{for } \mathbb{P}^2 = \tilde{\mathbb{P}}^2 \text{ a. e. } \omega \in \Omega^2.$$

*Proof of Lemma 6.4.* Applying distributional consistency of $\varphi$ and $\tilde{\varphi}$, we find, for all $A \in \mathcal{H}^2$,

$$\mathbb{P}^2(A) = \int \mathbb{P}^1(d\omega) \kappa(\omega, A) = \tilde{P}^2(A) \tag{4}$$

and thus $\mathbb{P}^2 = \tilde{\mathbb{P}}^2$. Next, we consider $A \in \mathcal{H}^2_{\rho(T^1)}$ and $S \subset \rho(T^1)$. Let us define

$$B = \{\omega \in \Omega^2 : K^2_S(\omega, A) < \tilde{K}^2_S(\omega, A)\}. \tag{5}$$

Since $K_S^2(\cdot, A)$ and $\tilde{K}_S^2(\cdot, A)$ are $\mathcal{H}_S^2$ measurable we find $B \in \mathcal{H}_S^2 \subset \mathcal{H}_{\rho(T^1)}^2$. Then the definition of causal spaces (see Definition 2.1) implies that

$$K_S^2(\omega', A \cap B) = \mathbf{1}_B(\omega')K_S^2(\omega', A). \tag{6}$$

Note that $A \cap B \in \mathcal{H}_{\rho(T^1)}^2$ so we can apply interventional consistency (2) for $\mathcal{C}^2$ and $\tilde{\mathcal{C}}^2$ and obtain for any $\omega$

$$
\begin{aligned}
\int \kappa(\omega, \mathrm{d}\omega')\mathbf{1}_B(\omega')K_S^2(\omega', A) &= \int \kappa(\omega, \mathrm{d}\omega')\mathbf{1}_B(\omega')K_S^2(\omega', A \cap B) = \int K_{\rho^{-1}(S)}^1(\omega, \mathrm{d}\omega')\kappa(\omega', A) \\
&= \int \kappa(\omega, \mathrm{d}\omega')\mathbf{1}_B(\omega')\tilde{K}_S^2(\omega', A \cap B) = \int \kappa(\omega, \mathrm{d}\omega')\mathbf{1}_B(\omega')\tilde{K}_S^2(\omega', A)
\end{aligned}
\tag{7}
$$

We integrate this relation with respect to $\mathbb{P}^1(\mathrm{d}\omega)$ and then apply distributional consistency and get

$$
\begin{aligned}
0 &= \int \mathbb{P}^1(\mathrm{d}\omega)\kappa(\omega, \mathrm{d}\omega')\mathbf{1}_B(\omega')(\tilde{K}_S^2(\omega', A) - K_S^2(\omega', A)) \\
&= \int \mathbb{P}^2(\mathrm{d}\omega')\mathbf{1}_B(\omega')(\tilde{K}_S^2(\omega', A) - K_S^2(\omega', A)) \\
&= \int_B \mathbb{P}^2(\mathrm{d}\omega')\,(\tilde{K}_S^2(\omega', A) - K_S^2(\omega', A)).
\end{aligned}
\tag{8}
$$

On $B$ the last term is strictly positive by definition. Thus we conclude that $\mathbb{P}^2(B) = 0$ and thus $\tilde{K}_S^2(\omega', A) \leq K_S^2(\omega', A)$ holds almost surely. The same reasoning implies the reverse bound and we conclude that $\mathbb{P}^2$ almost surely the relation

$$\tilde{K}_S^2(\omega', A) = K_S^2(\omega', A) \tag{9}$$

holds. $\qquad\square$

**Lemma 6.5** (Surjective Deterministic Maps). *Suppose $(f, \rho)$ is an admissible pair for the causal space $\mathcal{C}^1 = (\Omega^1, \mathcal{H}^1, \mathbb{P}^1, \mathbb{K}^1)$ to the measurable space $X^2 = (\Omega^2, \mathcal{H}^2)$ and assume that $\rho$ is surjective and $f : \Omega_1 \to \Omega_2$ measurable. If $f$ is surjective, there exists at most one causal space $\mathcal{C}^2 = (\Omega^2, \mathcal{H}^2, \mathbb{P}^2, \mathbb{K}^2)$ such that $(f, \rho) : \mathcal{C}^1 \to \mathcal{C}^2$ is a causal transformation.*

*If, in addition, $K_{\rho^{-1}(S^2)}^1(\cdot, A)$ is measurable with respect to $f^{-1}(\mathcal{H}_{S^2}^2)$ for all $A \in f^{-1}(\mathcal{H}^2)$ and all $S^2 \subset T^2$ then a unique causal space $\mathcal{C}^2$ exists such that $(f, \rho) : \mathcal{C}^1 \to \mathcal{C}^2$ is a causal transformation.*

*Proof of Lemma 6.5.* We first prove uniqueness. The relation $f_*\mathbb{P}^1 = \mathbb{P}^2$ that are necessarily true for deterministic maps (see Section 4) implies that $\mathbb{P}^2$ is predetermined. Moreover, we find that, by (3), for any $A \in \mathcal{H}^2$, $S \subset T^2$ and any $\omega \in \Omega^1$,

$$K_{\rho^{-1}(S)}^1(\omega, f^{-1}(A)) = K_S^2(f(\omega), A).$$

But since $f$ is surjective we conclude that due to interventional consistency $K_S^2(\omega', A)$ for $\omega' \in \Omega^2$ is unique.

To prove the existence we note that by assumption for fixed $A \in \mathcal{H}^2$ the function $K_{\rho^{-1}(S^2)}^1(\cdot, f^{-1}(A))$ is measurable with respect to $f^{-1}(\mathcal{H}_{S^2}^2)$. Now by the Factorisation Lemma (see Lemma A.1 in Appendix A) there is a measurable function $g : (\Omega^2, \mathcal{H}_{S^2}^2) \to \mathbb{R}$ such that

$$K_{\rho^{-1}(S^2)}^1(\omega, f^{-1}(A)) = g \circ f(\omega).$$

We define $K_{S^2}^2(\omega', A) = g(\omega')$. By surjectivity this defines $K_{S^2}^2$ everywhere and this defines a probability kernel because $g$ is measurable.

It remains to verify that the resulting $\mathcal{C}^2$ is indeed a causal space. Using interventional and distributional consistency we obtain

$$
\begin{aligned}
K_\emptyset^2(f(\omega), A) &= K_\emptyset^1(\omega, f^{-1}(A)) \\
&= \mathbb{P}^1(f^{-1}(A)) \\
&= f_*\mathbb{P}^1(A) \\
&= \mathbb{P}^2(A).
\end{aligned}
$$

This verifies the first property of causal spaces. For the second property we observe that for $A \in \mathcal{H}_{S^2}^2$, $S^1 = \pi^{-1}(S^2)$ using causal consistency

$$
\begin{aligned}
K_{S^2}^2(f(\omega), A \cap B) &= K_{S^1}^1(\omega, f^{-1}(A \cap B)) \\
&= K_{S^1}^1(\omega, f^{-1}(A) \cap f^{-1}(B)) \\
&= \mathbf{1}_{f^{-1}(A)}(\omega) K_{S^1}^1(\omega, f^{-1}(B)) \\
&= \mathbf{1}_A(f(\omega)) K_{S^2}^2(f(\omega), B).
\end{aligned}
$$

Here we used that $\mathcal{C}^1$ is a causal space and $f^{-1}(A) \in \mathcal{H}_{S^1}^1$. Thus, we conclude that we obtained a causal space $\mathcal{C}^2$. $\qquad \square$

**Lemma 6.6** (Perfect Abstraction on Intervened Spaces). *Let $\mathcal{C}^1 = (\Omega^1, \mathcal{H}^1, \mathbb{P}^1, \mathbb{K}^1)$ with $(\Omega^1, \mathcal{H}^1)$ a product with index set $T^1$ and $\mathcal{C}^2 = (\Omega^2, \mathcal{H}^2, \mathbb{P}^2, \mathbb{K}^2)$ with $(\Omega^2, \mathcal{H}^2)$ a product with index set $T^2$ be causal spaces, and let $(f, \rho) : \mathcal{C}^1 \to \mathcal{C}^2$ be a perfect abstraction.*

*Let $U^1 = \rho^{-1}(U^2) \subseteq T^1$ for some $U^2 \subseteq T^2$. Let $\mathbb{Q}^1$ be a probability measure on $(\Omega^1, \mathcal{H}_{U^1}^1)$ and $\mathbb{L}^1$ a causal mechanism on $(\Omega^1, \mathcal{H}_{U^1}^1, \mathbb{Q}^1)$. Suppose that, for all $S \subseteq U^2$ and $A \in \mathcal{H}^1$, the map $L_{\rho^{-1}(S)}^1(\cdot, A)$ is measurable with respect to $f^{-1}(\mathcal{H}_S^2)$, and consider the intervened causal spaces*

$$
\begin{aligned}
\mathcal{C}_I^1 &= (\Omega^1, \mathcal{H}^1, (\mathbb{P}^1)^{do(U^1, \mathbb{Q}^1)}, (\mathbb{K}^1)^{do(U^1, \mathbb{Q}^1, \mathbb{L}^1)}), \\
\mathcal{C}_I^2 &= (\Omega^2, \mathcal{H}^2, (\mathbb{P}^2)^{do(U^2, \mathbb{Q}^2)}, (\mathbb{K}^2)^{do(U^2, \mathbb{Q}^2, \mathbb{L}^2)}),
\end{aligned}
$$

*where $\mathbb{Q}^2 = f_* \mathbb{Q}^1$ and $\mathbb{L}^2$ is the unique family of kernels satisfying $L_S^2(f(\omega), A) = L_{\rho^{-1}(S)}^1(\omega, f^{-1}(A))$ for all $\omega \in \Omega^1$, $A \in \mathcal{H}^2$, and $S \subseteq U^2$. Then $(f, \rho) : \mathcal{C}_I^1 \to \mathcal{C}_I^2$ is a perfect abstraction.*

*Proof of Lemma 6.6.* First, we note that by Lemma 6.5 $\mathbb{L}^2$ exists and is unique. Thus, we need to verify distributional consistency and interventional consistency.

Let us first show $f_*(\mathbb{P}^1)^{do(U^1, \mathbb{Q}^1)} = (\mathbb{P}^2)^{do(U^2, \mathbb{Q}^2)}$. Since $(f, \rho)$ is a causal transformation (i.e., interventional consistency as in (2) holds), we find that, for $A \in \mathcal{H}^2$,

$$
\begin{aligned}
f_*(\mathbb{P}^1)^{do(U^1, \mathbb{Q}^1)}(A) &= \int \mathbb{Q}^1(d\omega) K_{U^1}^1(\omega, f^{-1}(A)) \\
&= \int \mathbb{Q}^1(d\omega) K_{U^2}^2(f(\omega), A) \\
&= \int (f_* \mathbb{Q}^1)(d\omega') K_{U^2}^2(\omega', A) \\
&= \int \mathbb{Q}^2(d\omega') K_{U^2}^2(\omega', A) \\
&= (\mathbb{P}^2)^{do(U^2, \mathbb{Q}^2)}(A).
\end{aligned}
$$

Here we used the change of variable for pushforward-measures.

Next, we show interventional consistency of $(f, \rho) : \mathcal{C}_I^1 \to \mathcal{C}_I^2$. For this, we introduce the shorthand $f_S = \pi_S \circ f$. Note that since $f_S$ is measurable with respect to $\mathcal{H}_{\rho^{-1}(S)}^1$ we can find $\tilde{f}_S$ such that $f_S(\omega) = \tilde{f}_S(\omega_{\rho^{-1}(S)})$. Note that, by the interventional consistency of $(f, \rho) : \mathcal{C}^1 \to \mathcal{C}^2$, we have

$$
K_{\rho^{-1}(S)}^1(\omega, f^{-1}(S)) = K_S^2(f(\omega), A) = K_S^2(\tilde{f}_S(\omega_S), A).
$$

We can now show for $A \in \mathcal{H}^2$ and $S^1 = \rho^{-1}(S^2)$ that

$$
\begin{aligned}
(K^1)_{S^1}^{(U^1, \mathbb{Q}^1, \mathbb{L}^1)}(\omega, f^{-1}(A)) &= \int L_{S^1 \cap U^1}^1(\omega_{S^1 \cap U^1}, d\omega_{U^1}') K_{S^1 \cup U^1}^1((\omega_{S^1 \setminus U^1}, \omega_{U^1}'), f^{-1}(A)) \\
&= \int L_{S^1 \cap U^1}^1(\omega_{S^1 \cap U^1}, d\omega_{U^1}') K_{S^2 \cup U^2}^2(\tilde{f}_{S^2 \setminus U^2}(\omega_{S^1 \setminus U^1}), \tilde{f}_{U^2}(\omega_{U^1}'), A) \\
&= \int \left( (\tilde{f}_{U^1})_*(L_{S^1 \cap U^1}^1(\omega_{S^1 \cap U^1}, \cdot)) \right)(d\overline{\omega}_{U^2}) K_{S^2 \cup U^2}^2(\tilde{f}_{S^2 \setminus U^2}(\omega_{S^1 \setminus U^1}), \overline{\omega}_{U^2}, A)
\end{aligned}
$$

$$= \int L^2_{S^2 \cap U^2}(f(\omega)_{S^2 \cap U^2}), \mathrm{d}\overline{\omega}_{U^2}) K^2_{S^2 \cup U^2}(f(\omega)_{S^2 \setminus U^2}), \overline{\omega}_{U^2}, A)$$

$$= (K^2)^{(U^2, \mathbb{Q}^2, \mathbb{L}^2)}_{S^2}(f(\omega), A).$$

This ends the proof. $\qquad\square$

## B.3   PROOFS FOR SECTION 6.2

**Lemma 6.7** (Perfect Abstraction and No Causal Effect). *Let $(f, \rho) : \mathcal{C}^1 \to \mathcal{C}^2$ be a perfect abstraction. Consider two sets $U^2, V^2 \subset T^2$ and denote $U^1 = \rho^{-1}(U^2)$ and $V^1 = \rho^{-1}(V^2)$. If $\mathcal{H}^1_{U^1}$ has no causal effect on $\mathcal{H}^1_{V^1}$ in $\mathcal{C}^1$, then $\mathcal{H}^2_{U^2}$ has no causal effect on $\mathcal{H}^2_{V^2}$ in $\mathcal{C}^2$.*

*Proof of Lemma 6.7.* Consider $A \in \mathcal{H}^2_{V^2}$ and any $S^2 \subset T^2$. Then for any $\omega' \in \Omega^2$ we find an $\omega \in \Omega^1$ such that $f(\omega) = \omega'$. Using interventional consistency and $f^{-1}(A) \in \mathcal{H}^1_{V^1}$ we conclude

$$\begin{aligned}
K^2_{S^2}(\omega', A) &= K^1_{\rho^{-1}(S^2)}(\omega, f^{-1}(A)) \\
&= K^1_{\rho^{-1}(S^2) \setminus \rho^{-1}(U^2)}(\omega, f^{-1}(A)) \\
&= K^1_{\rho^{-1}(S^2 \setminus U^2)}(\omega, f^{-1}(A)) \\
&= K^2_{S^2 \setminus U^2}(\omega', A).
\end{aligned}$$

This ends the proof. $\qquad\square$

**Lemma 6.8** (Perfect Abstraction and Active Causal Effects). *Let $(f, \rho) : \mathcal{C}^1 \to \mathcal{C}^2$ be a perfect abstraction. Consider two sets $U^2, V^2 \subset T^2$ and denote $U^1 = \rho^{-1}(U^2)$ and $V^1 = \rho^{-1}(V^2)$. Assume that $\mathcal{H}^2_{U^2}$ has an active causal effect on $\mathcal{H}^2_{V^2}$ in $\mathcal{C}^2$. Then $\mathcal{H}^1_{U^1}$ has an active causal effect on $\mathcal{H}^1_{V^1}$ in $\mathcal{C}^1$.*

*Proof of Lemma 6.8.* Since $\mathcal{H}^2_{U^2}$ has an active causal effect on $\mathcal{H}^2_{V^2}$ in $\mathcal{C}^2$, we find that there is an $\omega' \in \Omega^2$ and an $A \in \mathcal{H}^2_{V^1}$ such that

$$K^2_{U^2}(\omega', A) \neq \mathbb{P}^2(A).$$

By surjectivity there is $\omega \in \Omega^1$ such that $\omega' = f(\omega)$ and thus

$$\begin{aligned}
K^1_{U^1}(\omega, f^{-1}(A)) &= K^2_{U^2}(\omega', A) \\
&\neq \mathbb{P}^2(A) \\
&= \mathbb{P}^1(f^{-1}(A)).
\end{aligned}$$

The claim follows because $f^{-1}(A) \in \mathcal{H}^1_{U^1}$. $\qquad\square$

**Lemma 6.10** (Perfect Abstraction and Sources). *Let $(f, \rho) : \mathcal{C}^1 \to \mathcal{C}^2$ be a perfect abstraction. Consider two sets $U^2, V^2 \subset T^2$ and denote $U^1 = \rho^{-1}(U^2)$ and $V^1 = \rho^{-1}(V^2)$. Assume that $\mathcal{H}^1_{U^1}$ is a local source of $\mathcal{H}^1_{V^1}$ in $\mathcal{C}^1$. Then $\mathcal{H}^2_{U^2}$ is a local source of $\mathcal{H}^2_{V^2}$ in $\mathcal{C}^2$.*

*In particular, this implies that if $\mathcal{H}^1_{U^1}$ is a global source then $\mathcal{H}^2_{U^2}$ also is a global source.*

*Proof of Theorem 6.10.* Our goal is to show that $K^2_{U^2}(\cdot, A)$ is a version of the conditional probability $\mathbb{P}^2_{\mathcal{H}^2_{U^2}}(A)$ for $A \in \mathcal{H}^2_{V^2}$. It is sufficient to show that for all $B \in \mathcal{H}^2_{U^2}$ the following relation holds

$$\int \mathbb{P}^2(\mathrm{d}\omega') \mathbf{1}_A(\omega') \mathbf{1}_B(\omega') = \int \mathbb{P}^2(\mathrm{d}\omega') \mathbf{1}_B(\omega') K^2_{U^2}(\omega', A). \tag{10}$$

Using that $(f, \rho)$ is a perfect abstraction, $f^{-1}(A) \in \mathcal{H}^1_{V^1}$, $f^{-1}(B) \in \mathcal{H}^1_{U^1}$, and that $\mathcal{H}^1_{U^1}$ is a local source of $\mathcal{H}^1_{V^1}$ we find

$$\begin{aligned}
\int \mathbb{P}^2(\mathrm{d}\omega') \mathbf{1}_A(\omega') \mathbf{1}_B(\omega') &= \int f_* \mathbb{P}^1(\mathrm{d}\omega') \mathbf{1}_A(\omega') \mathbf{1}_B(\omega') \\
&= \int \mathbb{P}^1(\mathrm{d}\omega) \mathbf{1}_A(f(\omega)) \mathbf{1}_B(f(\omega))
\end{aligned}$$

$$= \int \mathbb{P}^1(\mathrm{d}\omega)\mathbf{1}_{f^{-1}(A)}(\omega)\mathbf{1}_{f^{-1}(B)}(\omega)$$

$$= \int \mathbb{P}^1(\mathrm{d}\omega)K_{U^1}^1(\omega, f^{-1}(A))\mathbf{1}_{f^{-1}(B)}(\omega)$$

$$= \int \mathbb{P}^1(\mathrm{d}\omega)K_{U^2}^2(f(\omega), A)\mathbf{1}_B(f(\omega))$$

$$= \int f_*\mathbb{P}^1(\mathrm{d}\omega')K_{U^2}^2(\omega', A)\mathbf{1}_B(\omega')$$

$$= \int \mathbb{P}^2(\mathrm{d}\omega')K_{U^2}^2(\omega', A)\mathbf{1}_B(\omega').$$

Thus we have shown that (10) holds and the proof is completed. $\qquad\square$

## C  EQUIVALENCE OF CAUSAL MECHANISM DEFINITIONS

Let us here briefly comment on the notation and equivalence of the notions $K_S(\omega, A)$ to $K_S(\omega_S, A)$. To clarify the equivalence let us recall the factorization lemma.

**Lemma C.1** (Factorization Lemma). *Let $T : \Omega \to \Omega'$ be a map and $(\Omega', \mathcal{A}')$ a measurable space. A function $f : \Omega \to [0, 1]$ is $\sigma(T)/\mathcal{B}([0, 1])$ measurable if and only if $f = g \circ T$ for some $\mathcal{A}'/\mathcal{B}([0, 1])$ measurable function $g : \Omega' \to [0, 1]$.*

We apply this where for $T$ we use the projection $\pi_S : (\Omega, \mathcal{H}) \to (\Omega_S, \mathcal{H}_S)$. Then, by the factorization lemma, a $\mathcal{H}_S$ measurable map $K_S(\cdot, A) : \Omega \to [0, 1]$ gives rise to a map $K_S'(\cdot, A) : \Omega_S \to [0, 1]$ such that

$$K_S(\omega, A) = K_S'(\pi_S\omega, A) = K_S'(\omega_S, A). \tag{11}$$

Moreover, this maps is unique because $\pi_S$ is surjective. On the other hand, give $K_S'(\cdot, A) : \Omega_S \to [0, 1]$ we can just define $K_S(\cdot, A) : \Omega \to [0, 1]$ by (11). For this reason $K_S(\omega, A)$ and $K_S(\omega_S, A)$ can be used equivalently when identified as explained here and with a slight abuse of notation we will use both depending on the context without indicating the different domains.