# OpenReview forum: "Products, Abstractions and Inclusions of Causal Spaces"
_auai.org/UAI/2024/Conference — UAI 2024 poster_

### Official Review · Reviewer_jtZs · 2024-03-13

**Q2-1 Originality-Novelty:** 3
**Q2-2 Correctness-Technical Quality:** 3
**Q2-5 Clarity Of Writing:** 3

**Q1 Summary And Contributions:**

The authors extend the theory of causal spaces which have recently been introduced by Park et al. [2023]. Causal spaces are a measure-theoretic axiomization of causality. Whereas in the Park et al. [2021] these spaces have been introduced and connections to SCMs and potential outcomes have been discussed, this paper
- introduces the notion of "product causal spaces" together with the notion of "causal independence" and
- introduces/discusses abstractions of and transformations between causal spaces in a more general way than, for example, Rubenstein et al. 2017.

**Q2-3 Extent To Which Claims Are Supported By Evidence:**

4: Excellent: all claims are supported by very convincing evidence (in the form of comprehensive experimental evaluation, rigorous mathematical proofs, detailed (pseudo-)code, precise references, well-motivated and realistic assumptions) and the authors deliver what they promise.

**Q2-4 Reproducibility:**

4: Excellent: key resources (e.g. proofs, code, data) are available and key details (e.g. proof sketches, experimental setup) are comprehensively described for competent researchers to confidently and easily reproduce the main results.

**Q3 Main Strengths:**

The theory of causal spaces seems very promising and this paper is a novel and a substantial addition to the original paper on causal spaces. Because causal spaces are much more general than Structural Causal Models or the Potential Outcome Framework, this paper should be interesting to anyone working in causality and having interest in a new theoretical framework. Moreover, this paper should be interested for people interested in abstractions of causal models.

The paper is generally well-written and very rigorous; basically all claims are supported by rigorous proofs. The mathematics seems correct.

**Q4 Main Weakness:**

General remark on causal spaces: Causal spaces so far primarily seem to be a theoretical construct. Especially given that the paper is quite measure-theory heavy, this paper may at least in the near future not be of too much interest for practitioners.

Weaknesses of this paper: The two subtopics, namely product causal spaces on the hand and transformations on the other hand, seem rather loosely connected. The reader might wonder why these two topics are presented next to each other. Moreover, the notion of causal independence appears without proper motivation or ("real-world") examples.
Furthermore, at this point, there are 1--2 minor mathematical issues that I think need further clarification (see below for details). There also seem to be a few typos (see below for details). Lastly, some equations and mathematical formulas here and there can be made more accessible (see below for details).

**Q5 Detailed Comments To The Authors:**

Remark 1: In the following, when I say original paper, I mean Park et al. 2023.

Remark 2: I do not expect the authors to respond to all of the following points. Most points (especially regarding language) are minor. I will highlight the points that I am most interested in by "(Interesting to me):". (If there are points that are not highlighted and you strongly disagree with them, I am also interested in them).

I will chronologically go through the paper:

- Abstract: According to the UAI template, you should avoid citations in the abstract, but if you do, you should essentially give the whole reference.
- Section 1.2: "In this paper, we write ...". Isn't this introduction of your notation better placed into Section 2?
- Section 2: For the trivial sigma-algebra, you write $\mathcal{H}_0=${$\emptyset, \mathcal{H}$}, but shouldn't it rather be $\mathcal{H}_0=${$\emptyset, \Omega$}?
- Section 2: You write: "... we use the shorthand $\pi_S = \pi_{ST}$", but shouldn't it rather be $\pi_S=\pi_{TS}$? If $S\supseteq U$ you also write $\pi_{SU}$ instead of $\pi_{US}$.
- (Interesting to me): Definition 2.1 (ii): In the original paper it was $K_S(\omega,\ldots)$; in this paper, you write $K_S(\omega_S,\ldots)$. Is there a reason for adapting this notation? I also think that in the remainder of the paper, you sometimes use the notation regarding this point from the original paper (see e.g., Def 4.2), and sometimes from the new paper (e.g., the example from the beginning).
- Definition 2.3: You write: " ... be a causal space, $U\in\mathcal{P}(T)$, $A\in\mathcal{H}$ ..." Here, two separate mathematical expressions appear directly after each other, which makes the sentence somewhat less clear to read. (This issue also occurs at other places in the paper, I will not separately mention all of them.) Maybe, at least on some occasions, it increases clarity if you include some additional words between two chained but separate mathematical expressions.
- Definition 3.1: I think you should comment on, maybe in a brief footnote, why the extension to $\mathcal{H}_1\otimes \mathcal{H}_2$ is unique. I think you do such an argument in Section B.1, but I think it would be good to briefly mention that in the main paper.
- General remarks to definition, lemmas etc.: I think it could increase the clarity of the paper if you include titles behind a definition, lemma etc.. For example, Definition 3.1 could have the title "Product Causal Spaces".
- General remarks to proofs: I think that at least for some lemmas, such as Lemma 3.2, it would help if you include a very rough proof idea in the main paper. For example, for Lemma 3.2 you could write that the proof is basically just using Def 3.1 to check the two conditions in Definition 2.1.
- (Interesting to me): Section 3.1: I think this section lacks some motivation and some examples (maybe even "real-world"). In what sense is the notion of "Causal independence" interesting? How does that notion occur when looking e.g. at Structural Causal Models? Maybe you could even have some "real-world" example such as the altitude-temperature example in the original paper. I see that Lemma 3.4 gives some motivation, but I think there could be more (in contrast to this, for Section 4 you have several examples).
- Section 3.1: You write that it is possible that two sigma-algebras can be causally independent but not probabilistically independent and vice versa. I think it would be nice if you include examples for these statements, maybe in the supplement or so.
- Section 3.1: You write: "... causally independent but not probabilistically independent, or vice versa. Again, independence is only defined for two $\sigma$-algebras". I think here you should write "... Again, causal independence is only defined for ..." to make clear that after the previous sentence you refer to "causal independence".
- Section 4: Last line on page 3: Isn't the measure $P^2$ missing in the space into which you map?
- (Interesting to me): Definition 4.1: Below the definition you write: "This definition captures the fact ...". Despite this explanation, I think I still do not entirely understand the motivation why $\kappa$ is defined as it is defined. Could you maybe give a brief explanation on this, maybe also in the paper?
- (Interesting to me): Definition 4.2: The notion of distributional consistency for me looks clear. I find the notion of interventional consistency more difficult to understand. Could you maybe intuitively describe in a few words what the terms on the left and right hand-side in equation (3) describe and where that assertion is motivated from?
- Below Definition 4.2: You write: "... indexed by $T^2\setminus \rho (S^1)$". Shouldn't it rather be $T^2\setminus \rho (T^1)$ instead? $S^1$ has not been introduced so far, only $T^1$ and $S$.
- Above Equation (4): You write "... with respect to $\mathcal{H}_{\rho^{-1}(S)}$. Isn't there an index missing here? So shouldn't it rather be $\mathcal{H}^1\_{\rho^{-1}(S)}$?
- Equation (10): Shouldn't be the second component of the mean be $x_2$ instead of $x_1$?
- Example 4.4: You write $\mathcal{C}^1\times \mathcal{C}^2$ to denote the product causal space. I think this notation has not been introduced in Definition 3.1.
- (Interesting to me): Equation (16): In the second line, you write $K_S(\omega,A_1)$. Isn't there an index missing here? Shouldn't it be $K^1_S(\omega,A_1)$ instead? I also do not entirely understand where the second equation is coming from. I see that $K^2_\emptyset(\omega_2,A_2)=\mathbb{P}^2(A_2)$, however, I do not see the rest. Moreover, I think it is good for clarity, if you include a further equation where you finally arrive at the equation you want to get for the product causal space $\mathcal{C}^1\times \mathcal{C}^2$.
- Example 4.5: You say that you denote the joint distribution of $(X, Y, M, H)$ by $P$. However later in the example, you seem to write $P_{XYMH}$.
- Example 4.5: I think you should write down what $\rho$ is. I agree that this fact is somewhat obvious, but I think writing it increases the reader's confidence that they understood this correctly when writing it out.
- Example 4.6: After equation (20) the example seems to continue, however, the font is not italic anymore. It seems that your example-environment is closed too early after equation (20).
- (Interesting to me):  4.6: I wonder about the following: Equation (20) for me seems like a strong restriction: I can write $\mathbb{P}^2(A)=\int 1_A(\omega)\mathbb{P}^2(d\omega).$ Thus, one should get that $\mathbb{P}^2(A)=\int (1_A(\omega)-K_{S^2}(\omega,A))\mathbb{P}^2(d\omega)=0$ which then implies that $1_A(\omega)=K_{S^2}(\omega,A)$ almost surely, or not? This would be quite a strong restriction, contrary as what you do in this example, or not?
- (Interesting to me): Definition 4.8: Could you give some intuition on why you additionally require that $f$ is surjective?
- Lemma 6.3: I think you do not need to introduce $[d]:=\{1,\ldots,d\},$ you have already done so in Section 1.2.
- (Interesting to me): Lemma 6.3: For me the term $P(X_i|\text{PA}_i)$ looks somewhat unrigorous. When writing about conditional distributions, you often condition on a particular realization of a variable, say you write $P\_{X|Y=y}$. Moreover, you write $P(X_1,\ldots,X_d)$ which is somewhat inconsistent with your notation in Example 4.5. Also, it is not entirely clear, whether you mean the product measure when writing $\Pi$. I am aware of this factorization result, however, more for probability densities or mass functions; I wonder, in how far and in what sense this factorization result generalizes to a general measure-theoretic setting and whether you are aware of a reference where this general measure-theoretic result is stated?
- (Interesting to me): Section 6.2: You write: "Note that Example 4.6 show that for general stochastic maps, we cannot expect ...". Could you explain in more detail in how far Example 4.6 shows what you then state?
- Between Lemma 6.6 and Lemma 6.7: You do not introduce the notion of an active causal effect. In contrast to the original paper, you only recall the notion of a causal effect. Maybe you also recall that notion in Section 2.
- Section 6.1: You write " ... lifting no causal effect form the target ...". I do not get that fragment. How can you lift something that is not there, in this case, "no causal effect"?
- Section 7: You write " ... but they have natural and important semantic interpretations. as causal ...". The full stop seems not to belong here.
- Section A: You write: "Recall that probability spaces are defined ...". Maybe you could write: "Recall that probability spaces $(\Omega,\mathcal{H},\mathbb{P})$ are defined ..." for better clarity.
- Section A: Point 3 in the definition of probability spaces: There seems to be a bracket missing after $(\Omega,\mathcal{H}$.
- Section B.1: You write: "Hence for this fixed pair $A_1$ and $A_2$ this $\omega$ ..." Isn't there an "and" missing in front of "this $\omega$"?
- Proof of Lemma 3.4: Maybe for people not so familiar with kernels you can write that the third equation follows by equation (29).
- Proof of Lemma 6.1: Below equation (35), you write $S^2=\rho^{-2}_2(S^3)$. Shouldn't it rather be $S^2=\rho^{-1}_2(S^3)$?
- Proof of Lemma 6.3: I think it is not optimal to both have $x_{S'}$ and $x'_S$ as both of these notations look very similar.
- (Interesting to me): Proof of Lemma 6.3: I do not entirely get the $\text{Mar}_{r\in R}$ notation. Where have you introduced that before?

### Literature:
- Park et al. [2023]: Junhyung Park, Simon Buchholz, Bernhard Schölkopf, and
Krikamol Muandet. A measure-theoretic axiomatisation
of causality. In Thirty-seventh Conference on Neural
Information Processing Systems, 2023.

- Rubenstein et al. [2017]: P Rubenstein, S Weichwald, S Bongers, J Mooij, D Janzing,
M Grosse-Wentrup, and B Schölkopf. Causal Consistency
of Structural Equation Models. In 33rd Conference on
Uncertainty in Artificial Intelligence (UAI 2017), pages
808–817. Curran Associates, Inc., 2017.

**Q9 Complying With Reviewing Instructions:**

Yes

---

> ### Author Rebuttal · Authors · 2024-04-05
>
> Let us first express our sincere gratitude to the reviewer for their extremely detailed review, going deep into the material and suggesting many improvements. We will update the manuscript to reflect all suggestions in the review. Our answers are short due to character limits, but we are happy to engage in further discussion.
>
> > (Interesting to me): In the original paper it was $K_S(\omega,\cdot)$; you write $K_S(\omega_S, \cdot)$ .
>
> Let us emphasize that the two notations are equivalent ways to define causal kernels. If the latter is used as a definition we can define $K_S(\omega, A)=\tilde{K}_S(\pi_S \omega, A)$. On the other hand, if $K_S(\omega,A)$ is used as a definition we obtain the existence of $\tilde{K}_S(\omega_S,A)$ by the factorization lemma. This is also used in the proofs of the original paper to exploit the $\mathcal{H}_S$ measurability.  We will ensure consistency with notation from prior work throughout.
>
> > (Interesting to me): Section 3.1: I think this section lacks some motivation and some examples (maybe even "real-world"). In what sense is the notion of "Causal independence" interesting?...
>
> As outlined in the general response, we will add further examples explaining this notion and the relation to SCMs. In the SCM setting, sets of variables that are disconnected from each other
> (no directed path from one set to the other) generate independent $\sigma$-algebras. See also the first part of our response to R. `2rXM`.
>
> > (Interesting to me): Definition 4.1: Below the definition you write: "This definition captures the fact ...".
>
> One difference between probability theory and causality seems to be that the latter requires the notion of variables (equivalently a product structure of the underlying space) that define entities that can be intervened upon. For a meaningful relation between two causal spaces, their interventions should be related, which requires some preservation of variables.
> Our definition states that every variable in the domain only influences a single variable in the target (capturing the abstraction of concepts).
>
> > (Interesting to me): ... I find the notion of interventional consistency more difficult to understand. ...
>
> The main idea is that intervention and transformation should commute (see Figure 1), i.e., certain interventions on the domain can be modelled by an intervention on the target. This is also similar to prior definitions in the literature.
>
> > (Interesting to me): Equation (16): ...
>
> You are correct about the missing index. We will add one more step to introduce $\kappa$ into the equation.
>
> > (Interesting to me): 4.6: I wonder about the following: Equation (20) for me seems like a strong restriction: ...
>
> Equation (20) itself is not a strong restriction that has several solutions (e.g, all measure preserving transformations, $1_A(\omega)$ and $K_{S^2}(\omega, A)=P(A)$). However, we overlooked the measurability condition of the causal kernels here
> and we apologize for this oversight. Therefore, we will remove this example and also
> the remark in Section 6.2 that you asked about below. When $\rho$ is not surjective the causal structure is not uniquely determined, see Example 4.4. Thank you very much for spotting this.
>
> > (Interesting to me): Definition 4.8: Could you give some intuition on why you additionally require that  is surjective?
>
> This condition is required, e.g., in Lemma 6.6 and 6.7 to relate properties of $\mathcal{C}^1$ and $\mathcal{C}^2$ and there it is a necessary assumption (otherwise we have, roughly speaking, no control over the part of $\mathcal{C}^2$ that is not in the image of $f$).
> The motivation to consider surjective maps is that we want to find a simpler, more concise description of a system. If $f$ is not surjective we could find a smaller space with domain $\Omega^2=f(\Omega^1)$. Note also that $f_\ast P^1$ is supported on the image of $f$, so the remaining part contains no probability mass.
>
> > (Interesting to me): Lemma 6.3: For me the term looks somewhat unrigorous...
>
> We agree that (26) is somewhat unrigorous and inconsistent with the rest of the paper.
> The factorization statement could be made rigorous under mild regularity assumptions ($E_i$ are Radon spaces and $\mathcal{H}_i$ is the Borel $\sigma$-algebra)
> using the theory of disintegration (a full treatment is in Chapter 45 in [1]). However, the conditional distributions are defined only almost everywhere.
>  We will make sure that the updated version contains a rigorous version of Eq. (38) under standard regularity assumptions for SCMs. Would this be OK for you?
>
> [1] Measure Theory, Volume 4, D Fremlin
>
> > (Interesting to me): Proof of Lemma 6.3: I do not entirely get the  notation.
>
> Unfortunately, we did not introduce the marginalization operation.
> For a measure $P$ on $\Omega = \bigtimes_{t\in T} E_t$ and  $R\subset T$ we define
>  $Mar_R P = (\pi_{T\setminus R}) \ast P$  as the pushforward of $P$ along the projection  $\pi_{T\setminus R}:\Omega\to \Omega_{T\setminus R}$.

---

### Official Review · Reviewer_wLis · 2024-03-19

**Q2-1 Originality-Novelty:** 3
**Q2-2 Correctness-Technical Quality:** 3
**Q2-5 Clarity Of Writing:** 4

**Q1 Summary And Contributions:**

The paper offers theoretical advancements building upon the causal spaces framework introduced by Park in 2023 in "A Measure-Theoretic Axiomatization of Causality", where causal spaces were constructed based on probability spaces enriched with causal kernels. In this work, the authors expand upon the concept of causal independence and introduce the notions of products of causal spaces and causal transformations between causal models. Multiple examples are provided to illustrate these new theoretical findings.

**Q2-3 Extent To Which Claims Are Supported By Evidence:**

3: Good: the main claims are supported by convincing evidence (in the form of adequate experimental evaluation, proofs, (pseudo-)code, references, assumptions).

**Q2-4 Reproducibility:**

3: Good: key resources (e.g. proofs, code, data) are available and key details (e.g. proofs, experimental setup) are sufficiently well-described for competent researchers to confidently reproduce the main results.

**Q3 Main Strengths:**

Valuable effort towards advancing the framework outlined by Park in 2023 in "A Measure-Theoretic Axiomatization of Causality".

**Q4 Main Weakness:**

* Lack of clear comparison with SCMs.
* It seems to me that the definition of causal independence lacks justification.

**Q5 Detailed Comments To The Authors:**

* Section 5: comparison with abstraction in SCM is not very clear. Maybe the clarity of this section can be improved with an example that  clearly shows the difference between an SCM and a causal space while highlighting the benefit of causal spaces in the context of abstractions.
* "connected components attract much more attention in the SCM community," Could you please elaborate on why the SCM community shows a greater interest in connected components? (I understand why it is true for abstractions and I would have understood the sentence if "connected components" is replaced by "causal graphs")?

* Can you give more details on how causally independent σ-algebras are analogous to connected components in graphical models?

**Q9 Complying With Reviewing Instructions:**

Yes

---

> ### Author Rebuttal · Authors · 2024-04-05
>
> We thank the reviewer for their positive review. We answer the specific questions below.
>
>
> > Section 5: comparison with abstraction in SCM is not very clear. Maybe the clarity of this section can be improved with an example that clearly shows the difference between an SCM and a causal space while highlighting the benefit of causal spaces in the context of abstractions.
>
> Causal spaces apply to more general settings that cannot be described by SCMs. E.g., stochastic processes indexed by the real numbers can be described by causal spaces but not through an SCM. We will add a suitable example of such a setting to future versions of the paper.
>
> > "connected components attract much more attention in the SCM community," Could you please elaborate on why the SCM community shows a greater interest in connected components? (I understand why it is true for abstractions and I would have understood the sentence if "connected components" is replaced by "causal graphs")?
>
> As you say, causal graphs are a building block of the SCM framework, and given a graphical structure, it is much more natural to consider properties such as connected components and abstraction. We believe that this is the reason why these concepts naturally attract more attention in the SCM community than the potential outcomes framework, which is not based on graphs.
>
>
> > Can you give more details on how causally independent σ-algebras are analogous to connected components in graphical models?
>
> We mean that the $\sigma$-algebras generated by the variables of connected components
>  are causally independent. We will clarify this in the paper.

---

### Official Review · Reviewer_2JEk · 2024-03-22

**Q2-1 Originality-Novelty:** 3
**Q2-2 Correctness-Technical Quality:** 3
**Q2-5 Clarity Of Writing:** 4

**Q1 Summary And Contributions:**

Consider any type of mathematical structure, like a topological space, metric space, etc.  There are many possible operations one can perform on those types structures, like forming products between them, or constructing subspaces, or etc.  Mathematical questions then arise about how the operations on the newly created structures (e.g., continuous functions on product topological spaces) are related to old ones and whether one has constructed the new space in the "right" way (e.g., whether the box or product topology is correct).  Similarly, mathematical questions arise about properties of structure-preserving functions from the old spaces into the newly constructed ones, and vice versa, and what the "right" type of structure-preserving function is (e.g., continuous functions for topological spaces, uniformly continuous functions for uniform spaces, etc.)

This paper fits into that broad mathematical tradition.  Park 2023 introduced a notion of a *causal space,* and this paper studies two possible operations on causal spaces:  (1) forming (finite) product spaces, and (2) forming abstractions.    This paper also studies transformations of those structures.  Some preliminary results are proven showing the definitions are "correct" (e.g., that a product of two causal spaces is itself a causal space); examples are given illustrating the abstract results, and in the cases

**Q2-3 Extent To Which Claims Are Supported By Evidence:**

4: Excellent: all claims are supported by very convincing evidence (in the form of comprehensive experimental evaluation, rigorous mathematical proofs, detailed (pseudo-)code, precise references, well-motivated and realistic assumptions) and the authors deliver what they promise.

**Q2-4 Reproducibility:**

4: Excellent: key resources (e.g. proofs, code, data) are available and key details (e.g. proof sketches, experimental setup) are comprehensively described for competent researchers to confidently and easily reproduce the main results.

**Q3 Main Strengths:**

The paper is clearly written, and the proofs appear to be correct.

**Q4 Main Weakness:**

As the author recognizes, there are no known empirical applications of the causal space framework, and it remains unclear whether the framework will ever yield empirical applications for which the SCM framework is inadequate.  The author sketches -- as Park does -- to the fact that the greater mathematical generality of causal spaces allows one to *represent* cyclical structures, for instance.  But the current paper -- and the framework generally -- hasn't been applied to make progress on any empirical problem.

**Q5 Detailed Comments To The Authors:**

See response to other questions.

**Q9 Complying With Reviewing Instructions:**

Yes

---

> ### Author Rebuttal · Authors · 2024-04-05
>
> We thank the reviewer for their review. We understand your general concern, and we addressed this in the general response. If any further questions come up, we are happy to address those.

---

### Official Review · Reviewer_2rXM · 2024-03-23

**Q2-1 Originality-Novelty:** 3
**Q2-2 Correctness-Technical Quality:** 3
**Q2-5 Clarity Of Writing:** 3

**Q1 Summary And Contributions:**

The paper takes a measure-theoretic approach to formulate the product and transformations of causal spaces. This leads to the interpretation of causal independence and abstractions in the language of causal spaces. The paper further shows some nice properties of causal effects when perfect abstractions are considered.

**Q2-3 Extent To Which Claims Are Supported By Evidence:**

3: Good: the main claims are supported by convincing evidence (in the form of adequate experimental evaluation, proofs, (pseudo-)code, references, assumptions).

**Q2-4 Reproducibility:**

3: Good: key resources (e.g. proofs, code, data) are available and key details (e.g. proofs, experimental setup) are sufficiently well-described for competent researchers to confidently reproduce the main results.

**Q3 Main Strengths:**

- The paper is clearly structured and written.
- The definitions of two mathematical operations -- product and transformation -- seem natural and intuitive.
- Intuitions are provided for most of the results. As someone who is not an expert in measure theory, I found the diagram (Figure 1) and examples quite helpful.

**Q4 Main Weakness:**

- While definitions 2.1-2.4 are taken from [Park et al. 2023], it will be helpful to add more explanations since these definitions are not easy to penetrate.
- The significance of causal independence and transformations needs to be clarified further.

**Q5 Detailed Comments To The Authors:**

1. Lemma 3.4 shows some connections between causal independence and causal effects. However, I'm unclear about the implication of these two results. Does it mean that knowing causal independence can help to identify causal effects?
2. In section 5, "They have to find a separate map w between the interventions themselves, whereas our map $\rho$ also determines the transformation of the causal kernels". Does this imply that [Rubenstein et al. 2017] allows more abstractions since it also considers other possible mappings?
3. Are there specific scenarios in which the inclusion (embedding) transformation is needed?
4. I don't think $w_S$ is defined in Definition 2.1 (ii).
5. Should it be $N((3x_1+x_2, \mathbf{x_2}), Id_2)$ in Example 4.3 Eq. (10)?

**Q9 Complying With Reviewing Instructions:**

Yes

---

> ### Author Rebuttal · Authors · 2024-04-05
>
> We thank the reviewer for their review and their helpful comments.
>
> > Lemma 3.4 shows some connections between causal independence and causal effects. However, I'm unclear about the implication of these two results. Does it mean that knowing causal independence can help to identify causal effects?
>
> In general, if we know that we have causal independence of two $\sigma$-algebras, then we foresee that this will help with identification of causal effects because we can rule out what is known as "interaction effects" in basic statistics (e.g. ANOVA), whereby the effect of one variable depends on another variable. However, this particular result is more of a sanity check of the definition.
>
> > In section 5, "They have to find a separate map w between the interventions themselves, whereas our map also determines the transformation of the causal kernels". Does this imply that [Rubenstein et al. 2017] allows more abstractions since it also considers other possible mappings?
>
> In principle the map $\omega$ provides additional flexibility. Note, however, that it is not clear in what context these transformations are meaningful. Also, we show that in certain settings (for pure abstraction, see Lemma 6.4) that the causal structure on the target is unique, so that $\omega$ is also unique. Let us also refer to the second comment to Question Q5 of R. `CNfE`:
> 'In fact, the reliance on a separate mapping for the interventions was one of the main criticisms of exact transformations that the work on abstractions starts out with.'
>
> > Are there specific scenarios in which the inclusion (embedding) transformation is needed?
>
>  Note that in probability theory, it is a natural operation to extend a probability space to accommodate additional variables. While this is rarely made explicit, it is implicitly used everywhere. Therefore, we think that it is useful to investigate similarly fundamental questions in the context of causality.
>
> > I don't think $w_S$ is defined in Definition 2.1 (ii).
>
> This is true, we will add this. We write $\omega_S$ for the vector $(\omega_s)_{s\in S}$.
>
> > Should it be .. in Example 4.3 Eq. (10)?
>
> Indeed, good catch!

---

### Official Review · Reviewer_CNfE · 2024-03-23

**Q2-1 Originality-Novelty:** 2
**Q2-2 Correctness-Technical Quality:** 3
**Q2-5 Clarity Of Writing:** 3

**Q1 Summary And Contributions:**

The paper takes the recent work of Park et al on causal spaces, and extends it by considering several relations between causal spaces. Concretely, it considers products of causal spaces and introduces the idea of causal dependence, based on their probabilistic counterparts. It then defines transformations of causal spaces, which include abstractions as well extensions into a richer space. They compare to existing work on abstractions, discuss a few examples, and provide several results on the properties of transformations.

**Q2-3 Extent To Which Claims Are Supported By Evidence:**

3: Good: the main claims are supported by convincing evidence (in the form of adequate experimental evaluation, proofs, (pseudo-)code, references, assumptions).

**Q2-4 Reproducibility:**

3: Good: key resources (e.g. proofs, code, data) are available and key details (e.g. proofs, experimental setup) are sufficiently well-described for competent researchers to confidently reproduce the main results.

**Q3 Main Strengths:**

The strength of the paper is very much a relative matter: if one views the contribution of Park et al on defining causal spaces as a valuable one, then the current paper is valuable as well, for it extends that framework in several interesting directions. I do believe that the  entirely probabilistic framework of causal spaces is a welcome contribution to the causal landscape, and thus I also consider the current paper to be of interest.

The formal definitions and results are mathematically rigorous, and it opens up interesting connections between probabilistic spaces and causal models, in particular along the dimensions of relating different causal models through transformations such as abstractions and embeddings.

**Q4 Main Weakness:**

The paper relies on prior knowledge of Park et al, giving only a brief introduction and explanation of causal spaces. Given that the original Park et al paper was already mathematically quite complex, the current one is even harder to process. I suspect that it is very hard to read as a self-contained work.

In fact, the entire formalism runs the danger of become so idiosyncratic and technical that only those who actually developed it will be able to truly understand and apply it. In all honesty, I did not consider all definitions in detail, for in the current form that it is presented (very dense and little explanation) and with my limited background in measure-theory, it would take me several days to fully comprehend every aspect. To be fair, perhaps one might say that this criticism applies to any complex novel formalism, so whether or not it is applicable depends on how popular the formalism eventually becomes, which is hard to predict, and thus we should give it the benefit of the doubt.

Furthermore, the paper introduces many novel definitions, leaving little space for providing intuitions and conceptual understanding of what exactly these definitions are about, let alone how they could be used in practice. The original paper contained more motivation for why one should be interested in moving away from SCMs, while this is somewhat lacking in the current paper.

**Q5 Detailed Comments To The Authors:**

As mentioned, I did not study all the definitions in detail, therefore I only have a few superficial clarificatory questions (and a few typos).

1: End of page 3: where is P^2?

2: Comment: the second aspect mentioned when comparing to exact transformation is in fact shared with abstractions: those also do not require a separate \omega. (In fact, the reliance on a separate mapping for the interventions was one of the main criticisms of exact transformations that the work on abstractions starts out with.)

3: The current discussion is highly theoretical. I realize space is restricted, but it would be good to add at least some motivation for the novel definitions by mentioning some applications. In particular, some practical examples where current definitions of transformations fall short and the novel definitions do not.

Typos:
Sec 4: "between of causal"
Sec 7: "interpretations. as"

**Q9 Complying With Reviewing Instructions:**

Yes

---

> ### Author Rebuttal · Authors · 2024-04-05
>
> We thank the reviewer for their helpful comments.
>
> > 1: End of page 3: where is $P^2$?
>
>  We will make this consistent. Thank you for pointing it out.
>
>  > 2: Comment: the second aspect mentioned when comparing to exact transformation is in fact shared with abstractions: those also do not require a separate \omega. (In fact, the reliance on a separate mapping for the interventions was one of the main criticisms of exact transformations that the work on abstractions starts out with.)
>
> Thank you very much for this addition, which we will add to the updated version of the paper.
>
>
>  > 3: The current discussion is highly theoretical. I realize space is restricted, but it would be good to add at least some motivation for the novel definitions by mentioning some applications. In particular, some practical examples where current definitions of transformations fall short and the novel definitions do not.
>
> As outlined in the general response, we will provide further simple examples using standard notions of causality to motivate and explain our definitions. We will also add an abstraction based on uncountable state space which is not covered by SCM theory.
>
>
> > Typos: Sec 4: "between of causal" Sec 7: "interpretations. as"
>
> We fixed those. Thank you for pointing them out!

---

### Meta-Review · Area_Chair_QZFn · 2024-04-16

I agree with the 5 reviewers (all of whom recommended acceptance) that the utility of the invention of "causal spaces" is yet to be decided. But is of theoretical interest nonetheless. Despite some of the weaknesses of the paper (insufficient comparison with SCM, no comparison with potential outcomes), the paper has many merits.